# A genetic screen in *Drosophila* uncovers the multifaceted properties of the NUP98-HOXA9 oncogene

Gwenaëlle Gavory[1]☯, Caroline Baril[1]☯, Gino Laberge[1]☯, Gawa Bidla[1], Surapong Koonpaew[1], Thomas Sonea[1], Guy Sauvageau[1,2], Marc Therrien[1,3]*

**1** Institute for Research in Immunology and Cancer, Université de Montréal, Montréal, Canada, **2** Département de médecine, Université de Montréal, Montréal, Canada, **3** Département de pathologie et de biologie cellulaire, Université de Montréal, Montréal, Canada

☯ These authors contributed equally to this work.
* marc.therrien@umontreal.ca

**Data Availability Statement:** All relevant data are within the manuscript and its Supporting Information files.

## Abstract

Acute myeloid leukemia (AML) underlies the uncontrolled accumulation of immature myeloid blasts. Several cytogenetic abnormalities have been associated with AML. Among these is the *NUP98-HOXA9* (*NA9*) translocation that fuses the Phe-Gly repeats of nucleoporin NUP98 to the homeodomain of the transcription factor HOXA9. The mechanisms enabling *NA9*-induced leukemia are poorly understood. Here, we conducted a genetic screen in *Drosophila* for modifiers of *NA9*. The screen uncovered 29 complementation groups, including genes with mammalian homologs known to impinge on NA9 activity. Markedly, the modifiers encompassed a diversity of functional categories, suggesting that NA9 perturbs multiple intracellular events. Unexpectedly, we discovered that NA9 promotes cell fate transdetermination and that this phenomenon is greatly influenced by *NA9* modifiers involved in epigenetic regulation. Together, our work reveals a network of genes functionally connected to NA9 that not only provides insights into its mechanism of action, but also represents potential therapeutic targets.

## Author summary

Acute myeloid leukemia or AML is a cancer of blood cells. Despite significant progress in recent years, a majority of afflicted individuals still succumbs to the disease. A variety of genetic defects have been associated to AML. Among these are chromosomal translocations, which entail the fusion of two genes, leading to the production of cancer-inducing chimeric proteins. A representative example is the NUP98-HOXA9 oncoprotein, which results from the fusion of the NUP98 and HOXA9 genes. The mechanism of action of NUP98-HOXA9 remains poorly understood. Given the evolutionarily conservation of NUP98 and HOXA9 as well as basic cellular processes across multicellular organisms, we took advantage of *Drosophila* fruit flies as a genetic tool to identify genes that impinge on the activity of human NUP98-HOXA9. Surprisingly, this approach identified a relatively large spectrum of conserved genes that engaged in functional interplay with NUP98-HOXA9, which indicated the pervasive effects that this oncogene has on basic cellular

**Funding:** G.G. was recipient of a doctoral studentship from Fonds de recherche du Québec - Santé (http://www.frqs.gouv.qc.ca/). M.T. holds a Canada Research Chair in Intracellular Signalling (https://www.chairs-chaires.gc.ca/). This work was supported by operating grants from the Canadian Institutes of Health Research (https://cihr-irsc.gc.ca/) (MOP93654) and the Leukemia & Lymphoma Society of Canada (https://www.llscanada.org/) to M.T and G.S. The funders had no role in study design, data collection and analysis, decision to publish, or preparation of the manuscript.

**Competing interests:** The authors have declared that no competing interests exist.

events. While some genes have been previously linked to NUP98-HOXA9, thus validating our experimental approach, several others are novel and as such represent potentially new avenues for therapeutic intervention.

## Introduction

Acute myeloid leukemia (AML) is among the most common and deadliest forms of leukemia affecting the ageing human population [1]. It is a clonal disease of hematopoietic stem cells characterized by the interruption of myeloid differentiation and the relentless proliferation of abnormal progenitors accumulating in bone marrow, blood and other tissues. Molecular lesions impinging on relatively few genes have been linked to the pathogenesis of AML [2]. *DNMT3A*, *FLT3*, *IDH1*, *IDH2*, and *NPM1* are among the most commonly mutated loci [3,4]. Recent advances in genomics is improving patient stratification, allowing for better therapeutic regimens [5,6]. However, the prognosis remains bleak and a deeper understanding of the underlying mechanistic causes of AML remains absolutely necessary to accelerate the development of effective therapies.

Homeobox (*HOX*) genes encode homeodomain-containing transcription factors that are the main regulators of mammalian development [7]. They also play essential roles in hematopoiesis throughout development and adult life [8]. Dysregulation of *HOX* gene expression in hematopoietic stem cells is closely associated to AML, which appears to be a common and cooperative event with driver mutations in genes such as *NPM1*, *FLT3* and *IDH1/2* [9–11]. A typical example is *HOXA9*, which is overexpressed in more than 50% of AML cases and has been defined as the most predictive marker of poor prognosis of AML [12,13]. Multiple mechanisms control the expression of *HOXA9* and their perturbations also lead to the development of AML. Among the clearest examples are the epigenetic regulators Mixed Lineage Leukemia (MLL; a histone H3K4 methyltransferase) and Polycomb Repressive Complex 2 (PRC2; harbors H3K27 methyltransferase activity) that positively and negatively regulate, respectively, the transcriptional activity *HOXA9* [14–16]. Aberrant activation of MLL by chromosomal translocations or inactivation of PRC2 subunits by loss-of-function mutations or silencing are conducive to AML onset and these genetic lesions are frequently accompanied by the upregulation of *HOXA9* expression [17]. Consistent with the relevance of HOXA9 in AML, its forced expression in murine bone marrow cells produces a preleukemic phase which, after a long latency, develops into full-fledged AML [18]. This last observation suggested early on the involvement of secondary collaborative events. These include the co-expression of the TALE (three amino-acid loop extension) family of co-factors, MEIS and PBX, which increase the DNA-binding affinity and specificity of HOX proteins [19–21] and significantly accelerate the onset of HOX-mediated AML [18,22].

Several recurrent chromosomal translocations also promote AML [23]. Among these, a diverse set of fusions involving nucleoporin genes have been detected, where *NUP98* is the most frequently affected gene [8]. The prototype chimera is *NUP98-HOXA9* (referred hereafter to *NA9*), which encodes the N-terminal Phe-Gly (FG)-rich repeat portion of NUP98 fused to the C-terminal portion of HOXA9 that comprises a PBX-Interacting Motif (PIM) and a DNA-binding homeodomain [24,25]. Mice transplanted with *NA9*-expressing bone marrow cells develop a myeloproliferative disease that ultimately progresses to AML after a long latency and is accompanied by an upregulation of the *HOX* loci [26]. As with *HOXA9*, *NA9*-induced leukemia is accelerated by co-expression of *MEIS1* [26].

Insights into NA9 activity was originally acquired by conducting structure-function analysis experiments in mammalian cells and *in vivo* mouse models of AML [27,28]. These studies suggested that NA9 acts as an aberrant transcription factor whereby the homeodomain binds to

DNA and the NUP98 moiety serves as a transcriptional activation domain. NUP98 FG repeats appear to influence transcription in part through physical interactions with the transcriptional co-activators CREB-binding protein (CBP) and p300, which are histone acetyltransferases (HAT), and with the transcriptional co-repressor HDAC1, a histone deacetylase [27,28]. NA9 also likely perturbs nucleocytoplasmic trafficking by sequestering the nuclear export factors RAE1 and Exportin1 (XPO1)/CRM1 by association with the GLEBS domain and FG repeats of the NUP98 moiety [29,30]. Conversely, chromatin-bound XPO1 was recently found to recruit NA9 to *HOX* genes and induce their expression [31]. MLL1 was shown to contribute to the oncogenic properties NA9 by recruiting NA9 to the *HOXA/B* locus via an interaction with the FG repeats of the NUP98 portion, thereby inducing *HOXA/B* gene expression [32,33]. Interestingly, a recent BioID screen conducted in the colon cancer cell line HCT-116, identified XPO1, RAE1, HDAC1 and MLL1 as proximal interactors of a NA9-BirA bait [34].

A variety of models have been used to characterize the molecular and cellular events underlying the leukemogenic activity of NA9 [26,28,35,36]. For example, we have recently shown that expression of human *NA9* in the hematopoietic organ of *Drosophila* larvae, called the lymph gland, triggers the premature differentiation of hemocyte progenitors followed by their proliferation [35]. This work revealed a need for the same functional elements as those originally delineated using mammalian models. We concluded that *Drosophila* could represent a relevant genetic system to identify molecular events impaired by NA9.

Since not required for viability or fertility, *Drosophila* eyes are particularly well suited for interrogating complex biological events by genetic means. Their use have indeed led to the discovery of numerous signaling mechanisms and developmental processes conserved across metazoans [37–39]. Emerging from an epithelium known as the eye-antennal imaginal disc, the *Drosophila* eye is a highly organized structure composed of about 800 photosensitive units called ommatidia [40]. Since closely related NUP98 and HOXA9 homologs are present in *Drosophila* (known as NUP98 and Abd-A/B, respectively), we reasoned that expression of human *NA9* in fly eyes could disrupt protein networks related to those perturbed in mammalian cells and thus produce phenotypes representative of NA9 function amenable to genetic screening.

Here, we show that expression of human NA9 during *Drosophila* eye development induces a phenotype that relies on the same functional elements as those originally defined in mammals. We exploited this system in a modifier screen to isolate genetic modulators of NA9 activity. This approach uncovered 29 complementation groups of mutations that dominantly alter the NA9 eye phenotype. Of these groups, three correspond to genes (*Rae1*, *emb* and *hth*) that have homologs in mammals (*RAE1*, *XPO1* and *MEIS1/2*) that have previously been reported to influence the leukemogenic activity of NA9 [26,29,31,41,42]. Interestingly, the screen uncovered evolutionarily-conserved genes encoding a variety of functions, such as chromatin remodeling, nuclear export, cell polarity, cytoskeletal organization and translation, suggesting that NA9 impinges on a multiplicity of cellular processes. Unexpectedly, a characterization of the *NA9*-induced eye phenotype revealed that it is largely based on the transdetermination of eye cells into wing cells and that several genetic modifiers of *NA9* influence this activity. Together, this study identifies NA9 as a disruptor of epigenetic regulation and unveils a large cohort of modifiers that might prove critical for its leukemia-inducing property.

## Results

### Expression of NA9 impedes eye development in an *exd* and *hth* dependent manner

To gain insights into NA9 function, we sought to identify modulators of its activity by conducting a genetic screen for dominant modifiers of a NA9-induced phenotype. We reasoned

that this approach should enable the isolation of heterozygous mutations in genes encoding factors that promote or, conversely, oppose NA9 activity. To facilitate the procedure, we looked for NA9-mediated developmental perturbations observable in adult flies that did not overtly compromise viability or fertility. To this end, we used an *eyeless (ey)-Gal4* driver to specifically target the expression of a *UAS-NA9* construct during eye development, which is a non-vital organ in laboratory conditions. This resulted in a "small eye" phenotype characterized by an expansion of the anterior dorsal and ventral head cuticles at the expense of the eye field (Fig 1C).

Major morphogens such as Hedgehog (Hh), Decapentaplegic (Dpp), and Wingless (Wg) control eye development, patterning and growth [40]. Given the impact of NA9 on eye shape and size, we evaluated its ability to alter the expression of these three morphogens (S1 Fig). In third instar eye discs, Hh is expressed posterior to the morphogenetic furrow (MF) in differentiating neurons, whereas Dpp is expressed within the MF. Hh is initially critical for MF initiation at the posterior margin and then both Hh and Dpp are required for MF progression across the eye field [43,44]. In contrast, Wg is expressed in the lateral margins, but anterior to the MF, where it antagonizes Dpp signaling and promotes head cuticle development at the expense of the eye [45]. *NA9*-expressing third instar eye discs under the *ey-Gal4* driver were consistently smaller than WT eye discs (S1 Fig), which correlates with the small eye phenotype seen in adult flies. Interestingly, they also exhibited higher Wg levels, especially at the dorsal margin (S1A and S1B Fig). Consistent with higher Wg levels, MF progression was markedly delayed in the dorsal compartment (S1C and S1D Fig). Interestingly, *dpp* expression was also reduced in the dorsal part of the MF, but not at the margin (S1C and S1D Fig). Finally, although the differentiating zone of NA9-expressing eye discs was reduced in size, it supported neuronal differentiation as revealed by expression of the neuronal marker ELAV (S1A and S1B Fig) and Hh levels were unaffected in this compartment (S1E and S1F Fig). Together, it appears that NA9 expression in eye discs augments Wg levels but reduces those of Dpp selectively in the dorsal compartment, and that these events likely cause MF progression delay leading to the small eye phenotype and head cuticle expansion.

We next assessed the specificity of the NA9 eye phenotype to determine whether it is related to known properties of the NA9 oncoprotein. Studies in mammals have shown that NA9 transformation and leukemia-promoting activity depends on the FG-rich repeats of NUP98 and the DNA-binding activity of the HOXA9 homeodomain [26,28,46]. The PBX-interacting motif (PIM) included within the C-terminal portion of HOXA9 was also shown to be required for the transforming activity of NA9 in NIH 3T3 mouse fibroblasts [28]. To assess the functional relevance of the various portions of NA9 leading to the eye phenotype, we tested the activity of a panel of *UAS-NA9* variants (Figs 1A and S2) driven to similar levels by *ey-Gal4*. As shown in Fig 1B–1F and quantified in Fig 1H, deletion of the HOXA9 moiety (NUP98$^{\Delta CT}$) or impairment of either the DNA-binding domain (NA9$^{HD}$) or the PIM motif (NA9$^{PIM}$) abolished the phenotype. In contrast, while deletion of the NUP98 moiety (HOXA9$^{\Delta NT}$) fully prevented the expansion of the anterior dorsal cuticle expansion, it still led to eye size reduction (Fig 1G), albeit less effectively than full-length NA9 (Fig 1H). Together, these results indicate that the NA9 eye phenotype depends on the known functional elements of the oncoprotein. They also suggest that the isolated HOXA9 moiety affects, but to a lesser degree, eye development independently of the N-terminal NUP98 portion.

HOXA9-dependent leukemia have been found to be influenced by the activity of the TALE transcription factors PBX and MEIS [18,22,47,48]. We wondered whether the eye phenotype also relied on the two respective TALE orthologues in flies, namely, Extradenticle (EXD) and Homothorax (HTH). To verify this possibility, we assessed the consequence of knocking down *exd* and *hth* transcripts by RNAi. As shown in Fig 1I and 1L, their individual depletion strongly

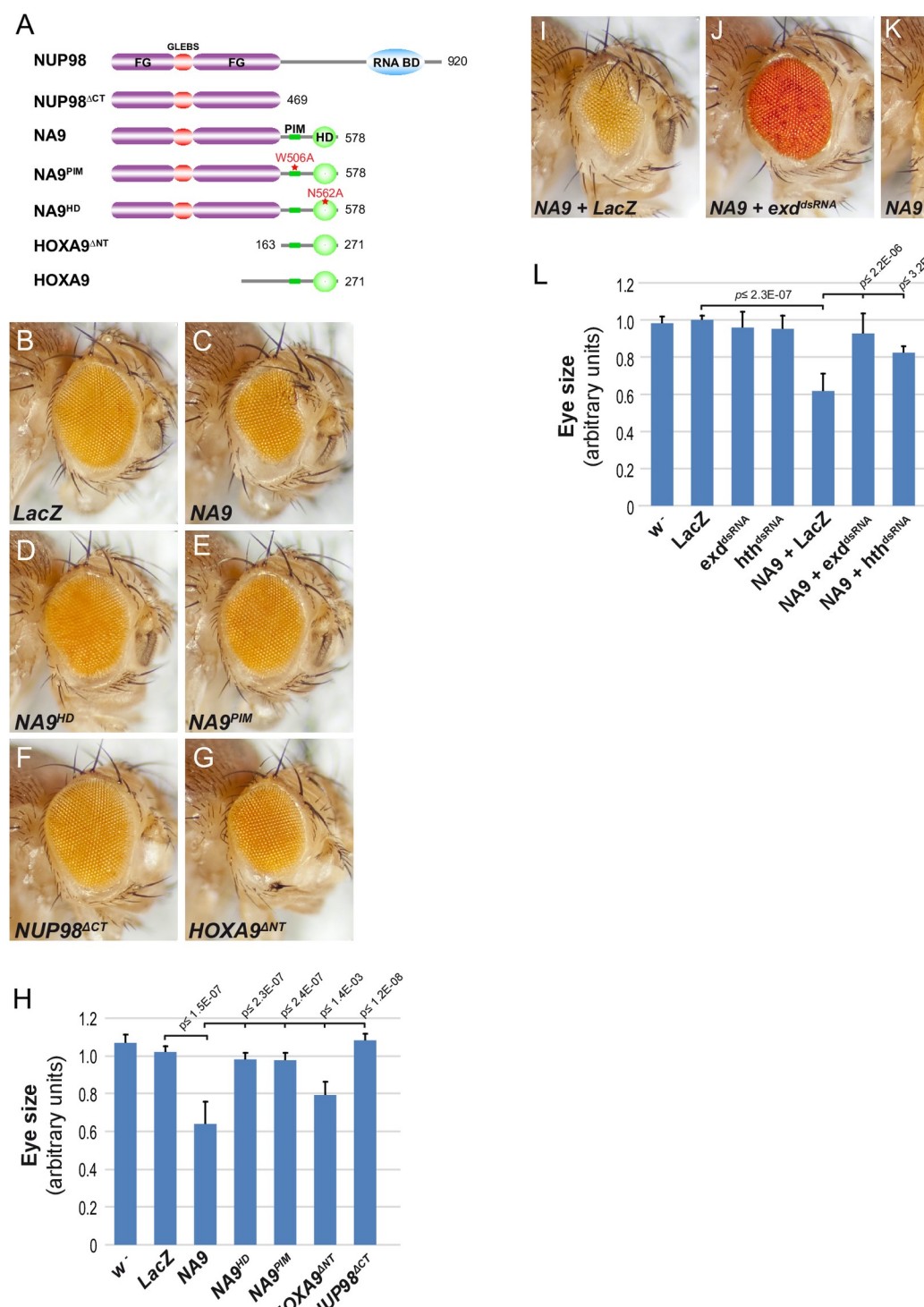

**Fig 1. NUP98-HOXA9 expression disrupts *Drosophila* eye development. (A)** Schematic representation of the NUP98-HOXA9 (NA9) variants used in this study. Full-length NUP98 and HOXA9 proteins are shown as reference. The W506A mutation in the PBX-Interaction Motif (PIM) abrogates PBX binding. The N562A mutation in the Homeodomain (HD) prevents DNA-binding. **(B-G and I-K)** Micrographs of representative adult *Drosophila* eyes expressing one copy of the indicated transgenes under the *UAS* promoter driven by *ey-Gal4*. The *LacZ* transgene is used as control. **(H and L)** Eye size quantification of the indicated genotypes. Statistical significance was assessed using a Student's *t* test. Posterior is to the left and dorsal is up. The same orientation is used for all adult eyes and eye imaginal discs shown throughout this study.

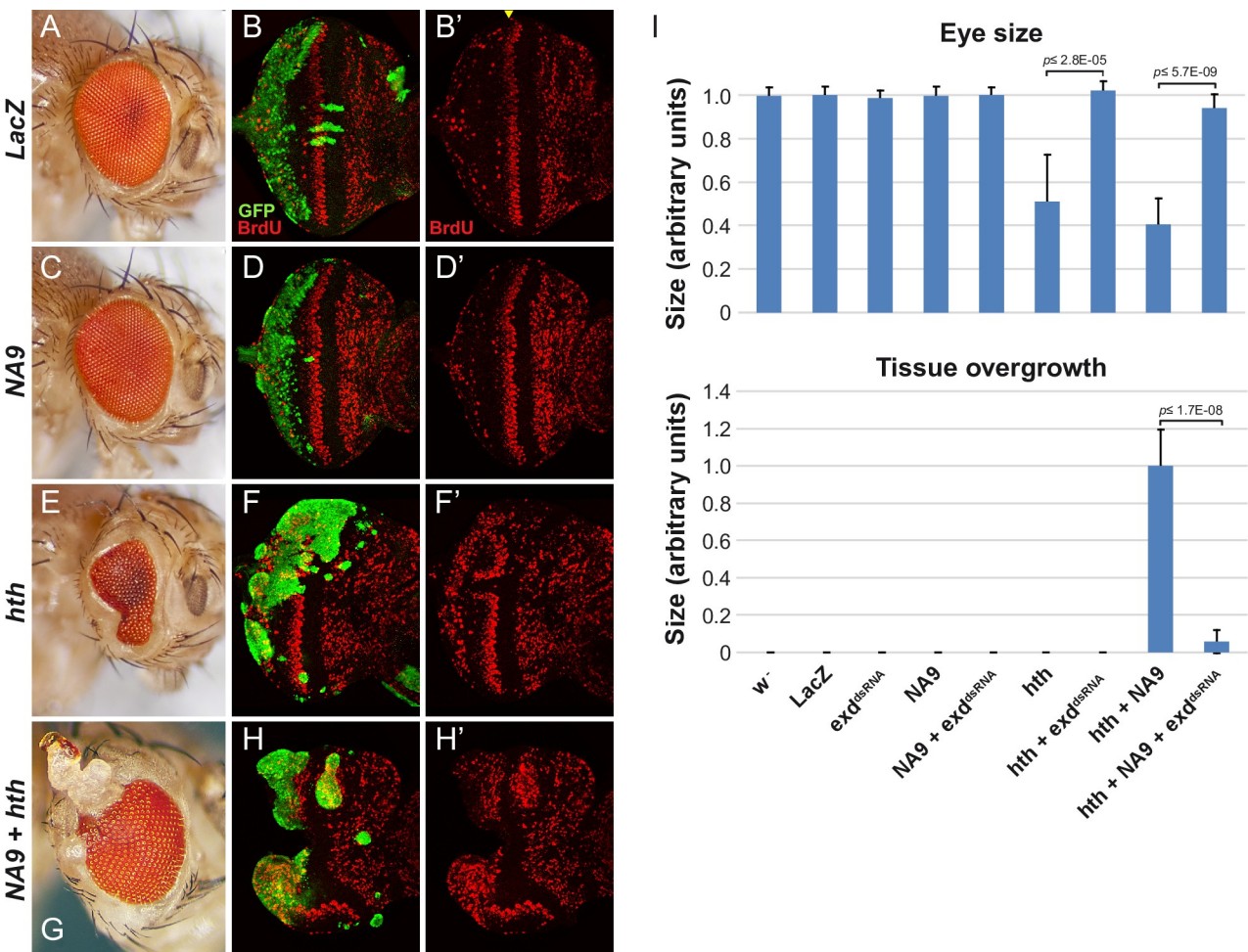

**Fig 2. NUP98-HOXA9 collaborates with HTH during *Drosophila* eye development.** FLP-out clones produced in the eye using *ey-FLP*; *Act5C > CD8 > Gal4*, *UAS-GFP* were analyzed at the adult and larval stages. **(A, C, E, G)** Micrographs of representative adult *Drosophila* eyes clonally expressing (A) *UAS-LacZ/+*, (B) *UAS-NA9/+*, (C) *UAS-GFP-hth*, and (D) *UAS-NA9/UAS-GFP-hth* as indicated to the left of each panel. **(B, D, F, H)** Third instar eye discs labeled with BrdU revealing cells in S phase. GFP staining marks the positions of clones expressing the transgenes indicated to the left of the panels. Yellow arrowhead indicates the position of the second mitotic wave. **(I)** Quantification of eye size and tissue overgrowth. Statistical significance was assessed using a Student's *t* test.

suppressed the NA9 eye phenotype, but had no effect on eye size when tested alone. These results demonstrate that NA9 activity in the eye also depends on the endogenous HOX co-factors EXD and HTH.

Another observation made in mammalian leukemia models is the ability of TALE co-factors to accelerate AML onset when co-expressed along with HOXA9 [18,22,48,49]. We therefore examined whether a similar collaboration could take place in the eye. The expression of *hth* was largely lethal when driven using *ey-Gal4*. To circumvent this limitation, we used the "Flp-out" system to clonally express *Gal4* and *UAS*-dependent constructs thereby confining expression to fewer cells [50]. Eye-specific expression was achieved using an *ey-flp* transgenic line that produces clones mostly in the posterior part of the eye disc. With this system, clonal expression of a single copy of the *NA9* transgene had no discernable impact on the adult eye (Fig 2C and 2I). As reported previously [51], significant overgrowth (assessed herein by clone size and BrdU incorporation) was observed in *hth*-expressing clones situated in the posterior part of the eye disc where differentiating cells are normally arrested in the G1 phase of the cell

cycle (compare Fig 2B and 2F). Although *hth*-driven hyperplasia was clearly visible in third instar eye discs, this never led to overgrowths in adult flies, but only to reduced eye size (Fig 2E and 2I) suggesting that *hth*-induced eye disc overgrowths are eliminated during metamorphosis. As expected, the *hth*-induced phenotype could be reverted by knocking down *exd* by RNAi (Fig 2I). In sharp contrast, while strong BrdU labelling was also observed in eye discs co-expressing *NA9* and *hth*, they consistently produced large cuticular overgrowths in the posterior region of adult eyes, which also could be prevented by depleting endogenous *exd* transcripts (Fig 2G and 2I). The emergence of cuticular overgrowths in adult eyes when NA9 and HTH are co-expressed suggests a cooperative event, reminiscent of the functional collaboration observed between NA9 and MEIS1 in mammalian systems. Taken together, the above findings recapitulate key observations made in mammalian cells and hence support the notion that the *NA9*-induced eye phenotype is a valid experimental paradigm to identify relevant modulators of NA9 activity.

## Identification of dominant modifiers of NA9 activity

The strength of the NA9 eye phenotype is dosage-sensitive (S3A–S3E Fig). This observation suggested its suitability as a readout to detect mutations in genes that influence NA9 activity. We thus took advantage of this phenotype to conduct an unbiased genetic screen in order to isolate heterozygous mutations acting as dominant modifiers of NA9, but that otherwise exhibit no phenotype on their own. The underlying principle is that a 2-fold reduction of a gene product that modulates NA9 activity, owing to a heterozygous mutation in that gene, should alter the sensitized NA9-mediated eye phenotype. Briefly, the screen was performed by crossing ethyl methanesulfonate (EMS)-mutagenized isogenic (chromosomes II and III) *white* (*w*)⁻ males to *w*⁻ females carrying an *ey-NA9* transgene inserted either on the *CyO* (*CNA9* flies) or *TM3* (*TNA9* flies) balancers. Approximately 100,000 adult F1 progeny were scored for suppressed or enhanced "small eye" phenotypes compared to the parental *NA9*-expressing stock. Only recessive lethal alleles linked to chromosome II or III were kept. Balanced lines were then grouped by complementation tests based on recessive lethality, which uncovered a total of 29 complementation groups of two or more alleles falling into 16 groups of *Suppressors of NA9* (*SN*) and 13 groups of *Enhancers of NA9* (*EN*) (Table 1). Representative examples are shown in Fig 3.

In addition to genuine modifiers, it is possible that mutations in genes that affect the activity of the *eyeless* promoter also alter the phenotype and thus be mistakenly identified as *NA9* modifiers. To identify such putative false positives, we crossed alleles for each group to flies expressing a *UAS-GFP* construct driven by *ey-Gal4*, which uses a similar *eyeless*-dependent expression system as the one employed in the screen. We then dissected third instar eye imaginal discs and quantified GFP fluorescence intensity as a proxy for transgene expression level (S4A Fig). Although significant GFP fluorescence variations was noted for some alleles, no single group of suppressors or enhancers consistently altered GFP expression (S4B and S4C Fig). Moreover, for those alleles that modified GFP expression, it did not correlate with their strength at modifying the *NA9* phenotype. These results suggest that *NA9* modifiers do not merely act by influencing NA9 transgene expression.

Since the *NA9* eye phenotype depended on endogenous EXD and HTH (Fig 1I–1L), we reasoned that *exd* and *hth* mutant alleles might have been recovered in the screen. However, the *exd* locus is on the X chromosome and thus putative *exd* alleles would not have been kept. With respect to *hth*, which is on the third chromosome, none of the groups of modifiers linked to that chromosome corresponded to *hth* mutations as assessed by complementation tests. Interestingly, multiple single hits have also been isolated in the screen and from those balanced

**Table 1. Groups of dominant modifiers of *NA9* on second and third chromosomes.**

| Groups | Genes | Cytological position | nb. of alleles | Human orthologs |
|---|---|---|---|---|
| *Chromosome II* | | | | |
| SN2-1 | grh | 54E10-F1 | 11 | GRHL1 |
| SN2-2 | mmp2 | 45F6-46A1 | 2 | MMP15 |
| SN2-3 | | 30E1-30E4 | 6 | |
| SN2-4 | | nd | 4 | |
| SN2-5 | l(2)gl | 21A5 | 4 | LLGL1 |
| SN2-6 | eIF3b | 54C3 | 4 | EIF3B |
| SN2-7 | stan | 47B6-B7 | 8 | CELSR2 |
| SN2-8 | AsnRS | 37C5 | 4 | NARS |
| SN2-9 | eIF3i | 25B5 | 2 | EIF3I |
| EN2-1 | emb | 29C1-C3 | 5 | XPO1 |
| EN2-2 | E(Pc) | 47F13-F14 | 2 | EPC1/2 |
| EN2-3 | | nd | 2 | |
| EN2-4 | | nd | 3 | |
| EN2-5 | | nd | 2 | |
| EN2-6 | | nd | 5 | |
| EN2-7 | ed | 24D4-D6 | 2 | KIRREL1/2/3 |
| EN2-8 | Rae1 | 57F6 | 3 | RAE1 |
| EN2-9 | CG6583 | 33D2 | 7 | Ly6-like |
| EN2-10 | | nd | 2 | |
| EN2-11 | toc | 23D1-D2 | 6 | MTUS2 |
| EN2-12 | shot | 50C6-C9 | 6 | DST/MACF1 |
| *Chromosome III* | | | | |
| SN3-1 | tara | 89B8-B9 | 2 | SERTAD1/2 |
| SN3-2 | | nd | 2 | |
| SN3-3 | | nd | 4 | |
| SN3-4 | | 77C6-77E1 | 4 | |
| SN3-5 | | 69B4-69C4 | 5 | |
| SN3-6 | | 90F4-90F4 | 2 | |
| SN3-7 | | nd | 2 | |
| EN3-1 | gpp | 83E7 | 3 | DOT1L |
| | hth * | 86C1-C3 | 1 | MEIS1 |

\* Single hit.

nd, not determined.

on the third chromosome, one single hit suppressor (*S-195*) was found to be allelic to *hth* (Table 1).

To confirm allelism, we isolated the genomic DNA of the *hth*[S-195] allele and sequenced the exons of the *hth* locus. This approach identified a single nucleotide change creating the K180N missense mutation in the *hth* open-reading frame. The K180 residue is evolutionarily conserved across MEIS-related proteins and falls within a highly conserved area (Figs 4A and S4D) that has previously been shown to act as a PBX/EXD-interaction region [52]. This result further confirms the role of HTH in mediating the *NA9* phenotype and provided unbiased evidence for the ability of the screen to identify functionally relevant modifiers.

To define the molecular identity of the groups of NA9 modifiers, we initially mapped them to specific chromosomal locations using the DrosDel deficiency kit collection covering

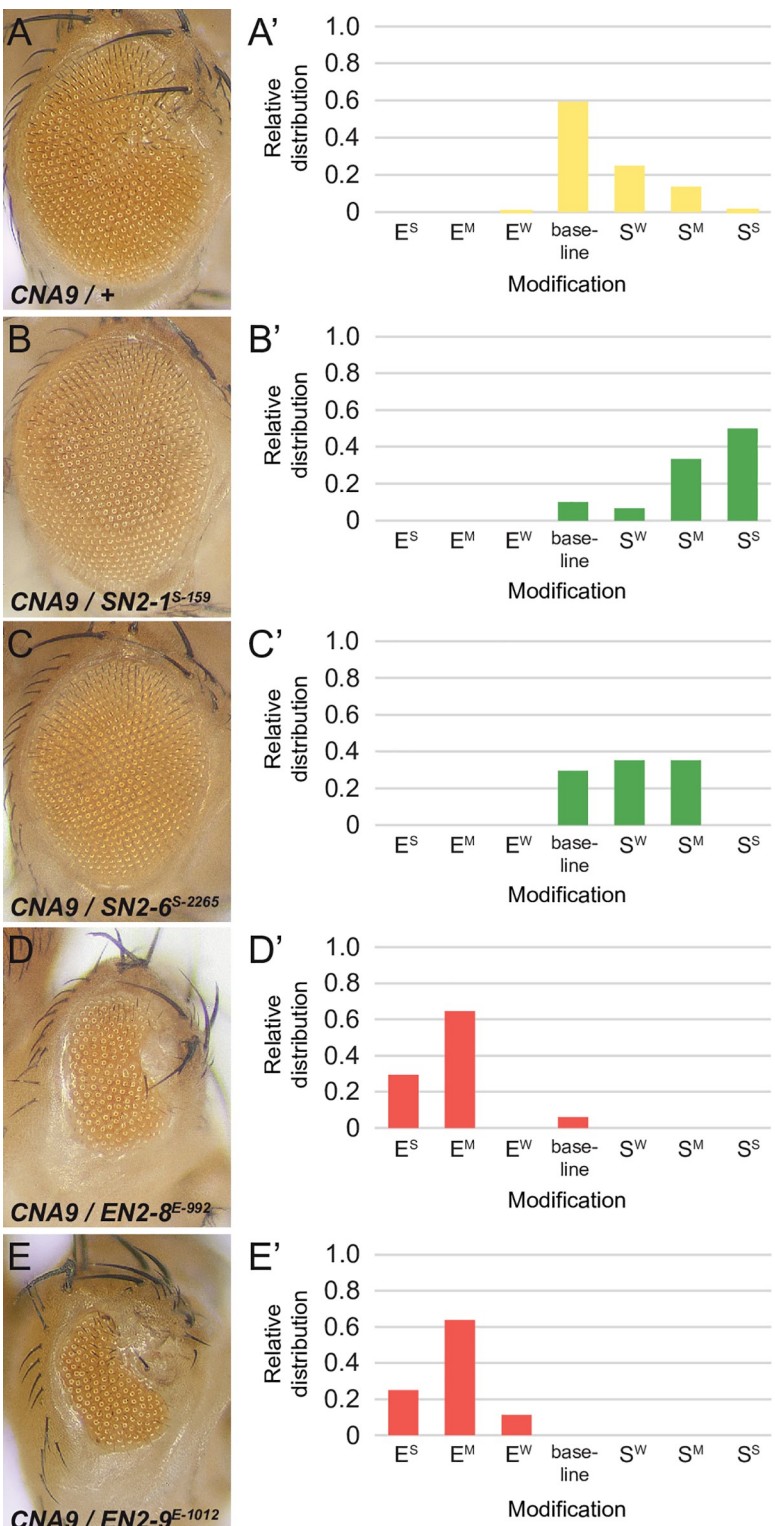

**Fig 3. Examples of modifiers recovered in the NA9 screen. (A-E)** Micrographs of representative adult *Drosophila* eyes of the indicated genotypes. Shown to the right are the corresponding relative distribution of eye modifications compared to the baseline phenotype induced by *NA9* expression. Note that the overall phenotype of the control *CNA9/+* genotype shows some variations mostly in the direction of mild suppression. Modifiers were scored as weak (W), medium (M) or strong (S) Enhancers (E) or Suppressors (S). At least 30 flies were scored for each cross presented.

chromosomes II and III. From this effort, a total of 20 groups were readily assigned a specific cytological position based on complementation tests (Table 1). Of these, 8 groups of suppressors and 8 groups of enhancers were linked to a specific gene following complementation tests with *P-element* insertions or other genetic lesions comprised within the breakpoints of the deficiencies (Table 1). Among the genes linked to suppressor groups, three are involved in translational control (*eIF3b*, *eIF3i*, and *AsnRS*), two in planar cell polarity (*l(2)gl* and *stan*), and the three others are respectively involved in epithelial morphogenesis (*grh*), tissue invasion (*mmp2*), and E2F transcriptional activity (*tara*).

With respect to the enhancers, we recovered two genes involved in nuclear export (*emb* and *Rae1*), two in epigenetic regulation (*E(Pc)* and *gpp*), while the remainders are respectively involved in cell adhesion (*ed*), cytoskeletal organization (*shot*), mitotic spindle organization (*toc*), and a Ly6 homolog (*CG6583*) of unknown function.

Markedly, mammalian counterparts for a number of the identified modifiers have previously been linked to NA9 function. For example, a recent study has reported that multiple subunits of the translation initiation factor eiF3 associates specifically with HOXA9, thus suggesting a role in translation [53]. The mammalian homolog of Grappa (GPP), DOT1L has been linked to MLL and NUP98-NSD1 fusion proteins in AML [54]. Perhaps the most striking cases are *embargoed* (*emb*) and *Rae1*, which respectively encode homologs of the nuclear exportin XPO1 and the RNA-export protein RAE1. These two factors have been shown to respectively interact physically with the FG repeats and the GLEBS domain of the NUP98 N-terminal region [29–31,41,42]. Some studies have also reported the influence of XPO1 and RAE1 on the leukemogenic activity of NA9 [31,42]. Moreover, NA9 was suggested to disrupt XPO1 activity and thereby increases the activity of transcription factors owing to their uncontrolled accumulation in the nucleus [29,41]. Expression of NA9 during eye development might have a similar effect and thus reducing the dose of EMB or RAE1 in eye cells owing to heterozygous mutations might exacerbate their impairment by NA9 expression. Consistent with this interpretation we found that low doses of selinexor, a selective XPO1 inhibitor, considerably enhanced the NA9 eye phenotype, but had no effect on WT eyes (Fig 4B–4E).

As further evidence of allelism between the modifier groups and specific genes, point mutations were uncovered to date in seven of the candidate genes thus confirming their identity (Fig 4A). A majority of the molecular lesions correspond to premature stop codons (11 out of 15), which suggests that most are loss-of-function alleles. To verify this point, we tested the effect of depleting by RNAi several of the inferred gene products on the *NA9* phenotype. While most dsRNA lines displayed no or mild eye phenotypes when expressed on their own (S5 Fig), they all modified the *NA9* phenotype as expected (Fig 5). Specifically, depletion of two suppressor genes, *grh* and *stan*, potently attenuated the *NA9* eye phenotype (Fig 5A–5C). In contrast, depletion of seven enhancer genes, *emb*, *Rae1*, *E(Pc)*, *ed*, *CG6583*, *shot*, and *gpp*, consistently intensified the *NA9* phenotype (compare Fig 5A to 5D–5J). RNAi-mediated knockdowns were validated either using immunostaining or qPCR analysis (S6 Fig). These findings support the notion that most of the recovered alleles are loss-of-functions. Furthermore, they provide an independent demonstration that the identified loci are *bona fide NA9* modifiers.

## NA9 expression induces eye-to-wing transdetermination

When tested by RNAi, the enhancers assessed above varied in their ability to modify the *NA9* phenotype. For instance, *emb* and *Rae1* knockdowns severely reduced the eye size and the head capsule (Fig 5D and 5E). In contrast, knockdowns of *ed*, *CG6583*, *shot*, and *gpp* led to a reproducible cuticular overgrowth in the dorsal-anterior region of the eye (Fig 5F–5I). Lastly,

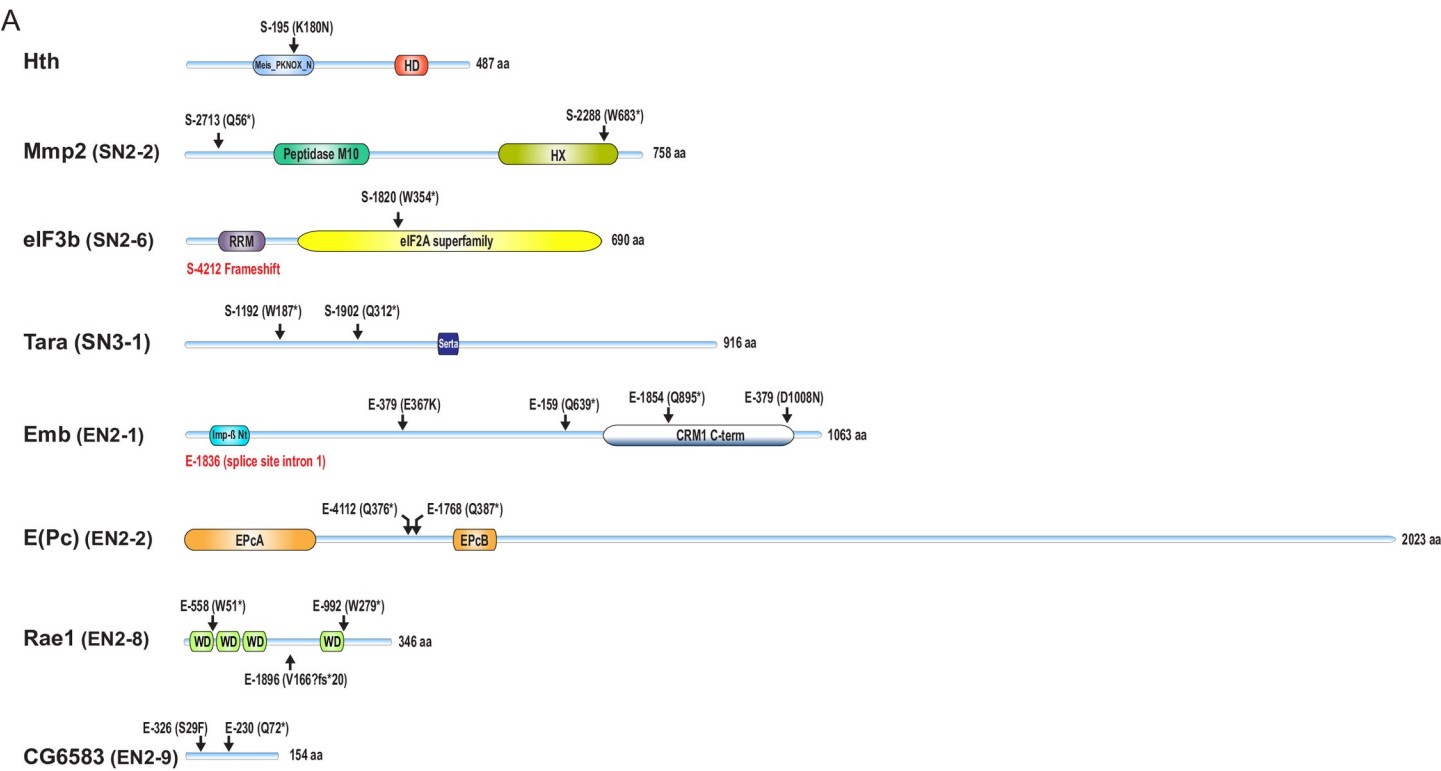

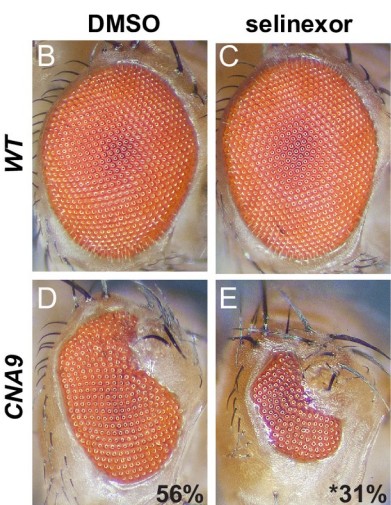

**Fig 4. Molecular lesions associated with eight modifiers of NA9. (A)** Schematic representations of proteins encoded by loci identified in the screen. Domain compositions were defined according to the Conserved Domains tool at the National Center for Biotechnology Information (NCBI)). The position of amino acid changes for the indicated sequenced alleles is shown on top of each protein schematics. Asterisks denote STOP codons. **(B-E)** Micrographs of adult eyes of the following genotypes: (B, C) Oregon R flies denoted as *WT* and (D, E) *CNA9/+* denoted as *CNA9*. Developing flies were treated with DMSO (B and D) or selinexor (C and E). Mean eye size (expressed as percent compared to *WT*), is indicated at the bottom right of *CNA9* panels. The experiment was conducted in triplicate and at least five eyes have been analyzed per condition. Statistical significance (*; $p \leq 4.2E\text{-}07$) of eye size difference between DMSO versus selinexor-treated *CNA9* flies (D, E) was assessed using a Student's *t* test.

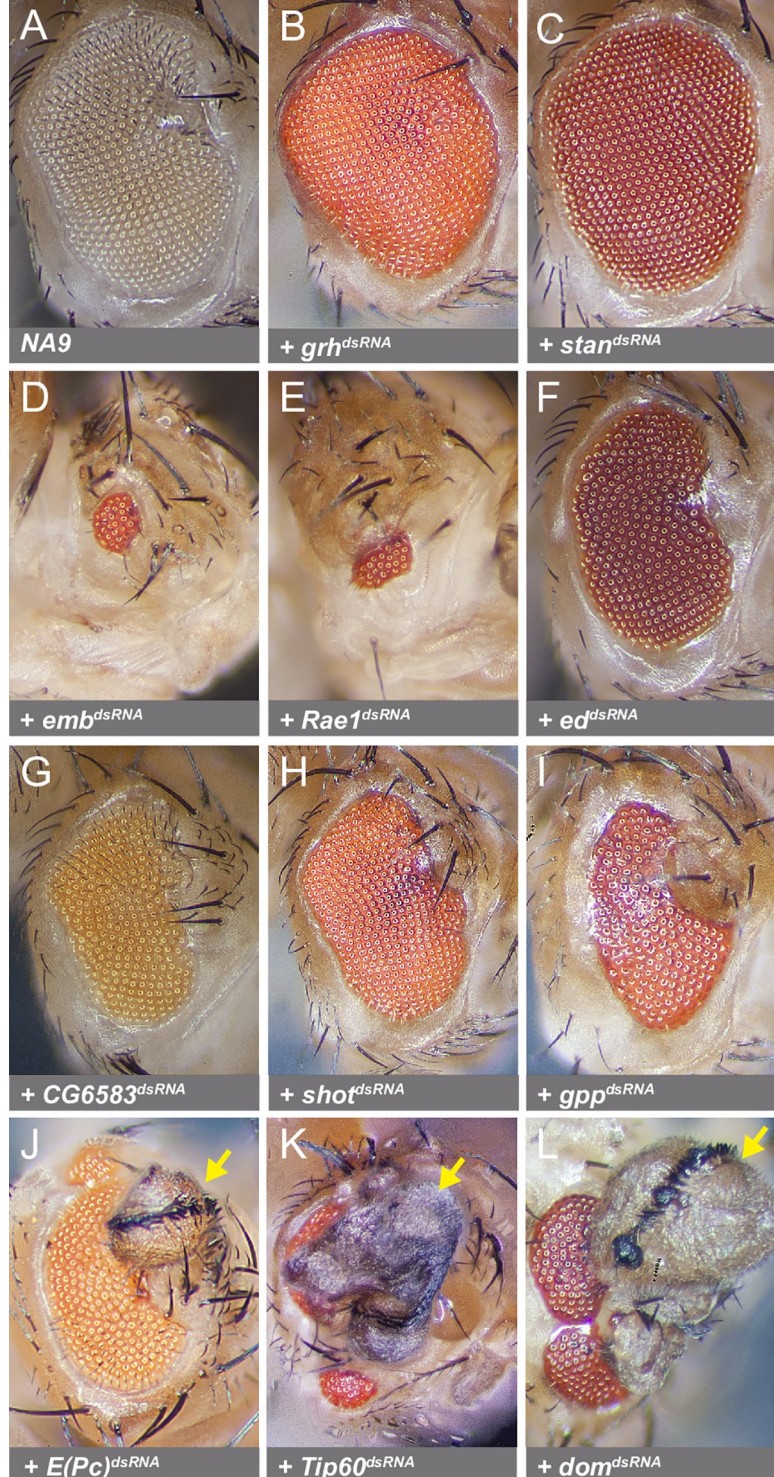

**Fig 5. Knockdown of candidate genes by RNAi modulates the *NA9* phenotype in a manner similar to mutant alleles.** Micrographs of adult *Drosophila* eyes of the following genotypes: **(A)** *ey-Gal4, UAS-NA9/+ (NA9).* **(B-L)** *ey-Gal4, UAS-NA9/UAS-dsRNA* lines as indicated. Yellow arrows point to wing-like structures emerging mostly from the anterior portion of adult eyes.

*E(Pc)* depletion markedly induced the formation of wing-like structures in the dorsal-anterior part of the eye (Fig 5J). The diverse enhanced phenotypes suggest that *NA9* expression perturbs distinct intracellular mechanisms.

We were intrigued by the ability of NA9 to induce the formation of wing-like structures in the eye upon *E(Pc)* depletion. Enhancer of Polycomb or *E(Pc)* is part of a multi-subunit complex called the NuA4/Tip60 histone acetyltransferase (HAT) complex that acts as a histone H2A and H4 acetyltransferase to control gene expression and genome stability through chromatin remodeling [55]. To determine whether the effect seen by *E(Pc)* depletion reflects its role through the Tip60 complex, we knocked down other Tip60 subunits in the presence of *NA9* expression. As shown in Fig 5K and 5L, depletion of *Tip60* or *domino* (*dom*) produced ectopic wing structures in *NA9*-expressing eyes with high penetrance (quantifications shown in Table 2). However, they did not produce wing material when expressed alone (S5K and S5L Fig). These findings suggest that *NA9* expression can reprogram eye development towards the wing fate and that the Tip60 complex works against this phenomenon.

During wing development, the transcriptional unit formed by Vestigial (Vg) and Scalloped (Sd) induces several wing-specific genes that are essential for wing identity and development [56]. Although not expressed during normal eye development (Figs 6A and S7A), Vg and Sd have previously been shown to be upregulated in eye discs undergoing eye-to-wing transdetermination [57,58]. Consistent with an ability to induce ectopic wing material in the eye, *NA9*-expressing third instar eye discs exhibited Vg and Sd expression with high penetrance at the margin of the anterior dorsal region (Figs 6H and S7B and Table 2). Moreover, *E(Pc)* loss-of-functions enhanced *NA9*-induced Vg expression (Fig 6J and 6L and Table 2), but had no effect on their own (Fig 6D and 6F and Table 2). Of note, the eye disc areas exhibiting Vg expression were systematically larger upon *E(Pc)* knockdown compared to the heterozygous *E(Pc)* alleles (Fig 6J and 6L), which could explain why wing-like structures in adult eyes were observed only in conditions of RNAi-mediated *E(Pc)* depletion (Table 2).

*NA9* expression readily leads to *vg* upregulation in the eye, thus suggesting its intrinsic ability to reprogram cell fate. We therefore asked if increasing *NA9* expression levels would result in greater expression of wing-determining genes and hence suffice to promote ectopic wing formation. We previously reported a second *UAS-NA9* transgenic line referred to as *line 6* that expressed the *NA9* transgene at levels ~ 6-fold higher compared to the reference line (line 5) used herein [35]. Remarkably, expression of *NA9^line6^* with the *ey-Gal4* driver resulted in high ectopic levels of Sd and Vg in third instar larval eye discs (S7C and S8F Figs), which were accompanied by wing-like structures in the eyes of 43% of the adult progeny (S8E Fig and Table 2). These results demonstrate the autonomous ability of NA9 to transform eye cells towards the wing fate.

As another evidence of the ability of NA9 to alter cell fate, we noted that *ey-Gal4* activity was consistently abrogated in the dorsal anterior part of third instar eye discs (S1B' and S8D' Figs) and this correlated with the extinction of expression of retinal determination factors such as Eyes Absent (Eya; [59]) in the same compartment (compare Fig 6B' to 6H'). To confirm that the dorsal anterior compartment did express the *NA9* transgene at the onset of *ey-Gal4* expression, we conducted a lineage tracing experiment using the G-TRACE system [60]. Compared to WT, the "real-time" activity of *ey-Gal4* was restricted to the posterior and ventral side of the eye disc in the presence of *NA9* expression (S9A and S9B Fig), whereas lineage expression of *ey-Gal4* was clearly distributed across the disc irrespective of the presence of *NA9* (S9A' and S9B' Fig). We conclude that the *NA9* transgene under the eye-specific *ey-Gal4* driver was indeed active at an earlier time point in the dorsal anterior part of the developing eye field, but its expression was silenced in this area of third instar eye discs upon adopting a wing fate.

**Table 2. Quantification of *NA9*-induced eye-to-wing transdetermination.**

| Lines tested | ey-Gal4 | | | | ey-Gal4, UAS-NA9 | | | |
|---|---|---|---|---|---|---|---|---|
| | Vg⁺ eye discs (%) | N | Wing tissues in adult eyes (%) | N | Vg⁺ eye discs (%) | N | Wing tissues in adult eyes (%) | N |
| *w*^1118^ | 0 | 263 | 0 | 1291 | 64 | 297 | 0 | 2603 |
| *E(Pc)*^E-4112^ | 0 | 79 | 0 | 435 | 99 | 107 | 0 | 305 |
| *E(Pc)*^E-1768^ | 0 | 88 | 0 | 379 | 100 | 66 | 0 | 178 |
| *UAS-E(Pc)*^dsRNA^ | 0 | 77 | 0 | 324 | 100 | 104 | 59 | 384 |
| *UAS-Tip60*^dsRNA^ | nd | nd | nd | nd | nd | nd | 42 | 52 |
| *UAS-dom*^dsRNA^ | nd | nd | nd | nd | nd | nd | 58 | 52 |
| *UAS-NA9*^line 6^ | 100 | 69 | 43 | 325 | nd | nd | nd | nd |
| *nej*^G0350^ | 0 | 34 | 0 | 12 | 92 | 40 | 19 | 155 |
| *nej*^P^ | 0 | 44 | 0 | 129 | 100 | 35 | 18 | 180 |
| *nej*^3^ | 0 | 18 | 0 | 41 | 100 | 34 | 61 | 92 |
| *UAS-nej-V5* | 0 | 28 | 0 | 95 | 5 | 43 | 0 | 99 |
| *UAS-HDAC1*^dsRNA^ | 0 | 23 | 0 | 247 | 17 | 51 | 0 | 73 |
| *UAS-HDAC1-V5* | 0 | 19 | 0 | 249 | 98 | 37 | 0 | 360 |
| *emb*^E-379^ | 0 | 17 | 0 | 70 | 89 | 53 | 0 | 299 |
| *emb*^E-1836^ | nd | nd | nd | nd | 96 | 47 | 8 | 138 |
| *gpp*^E-1750^ | 0 | 22 | 0 | 50 | 94 | 34 | 0 | 65 |
| *gpp*^E-3000^ | nd | nd | nd | nd | 93 | 45 | 0 | 13 |
| *UAS-gpp*^dsRNA^ | 0 | 19 | 0 | 70 | 98 | 42 | 0 | 266 |
| *Rae1*^E-558^ | nd | nd | 0 | 100 | 100 | 35 | 3 | 234 |
| *Rae1*^E-992^ | 0 | 18 | 0 | 157 | 98 | 60 | 1 | 237 |
| *Rae1*^E-1896^ | nd | nd | 0 | 100 | 100 | 39 | 0 | 137 |
| *CG6583*^E-326^ | nd | nd | nd | nd | 100 | 35 | 0 | 65 |
| *CG6583*^E-1448^ | nd | nd | nd | nd | 100 | 31 | 3 | 138 |

N, number of scored eye discs or flies.

nd, not determined.

## Eye-to-wing cell fate change induced by NA9 is linked to CBP and HDAC1

In addition to RAE1 and XPO1, NA9 physically associates through its NUP98 portion to the acetyltransferases CBP/p300 as well as to the histone deacetylase HDAC1 in mammalian cells [27,28,61]. Since these enzymes play a role in transcriptional control, we investigated whether the *Drosophila* counterparts might contribute to the transdetermination activity of NA9. *Drosophila* harbors a single CBP/p300 homolog called Nejire (Nej; [62]). No mutations in the *nej* locus have been recovered in our genetic screen as the gene is on the X chromosome. We tested whether a genetic interaction with *NA9* could nonetheless be detected using *nej* loss-of-function alleles from the Bloomington stock center. Strikingly, the three alleles tested as heterozygotes strongly enhanced NA9 transdetermination activity leading to ectopic wing formation in a considerable proportion of adult eyes (Fig 7C and Table 2). In agreement with this observation, Vg expression was strongly enhanced in *NA9*-expressing third instar eye discs (Fig 7D and Table 2). Importantly, *nej* heterozygous flies did not exhibit eye-to-wing transformation on their own nor did they show Vg expression in third instar eye discs (S10D Fig and Table 2). Conversely, expression of a *nej* transgene using *ey-Gal4* strongly suppressed *NA9* eye phenotype and Vg expression (Fig 7E and 7F and Table 2). It is interesting to note that, compared to other mutant loci tested thus far, *nej* heterozygous mutations are the only ones enhancing NA9 transdetermination activity up to the point of producing a large proportion of

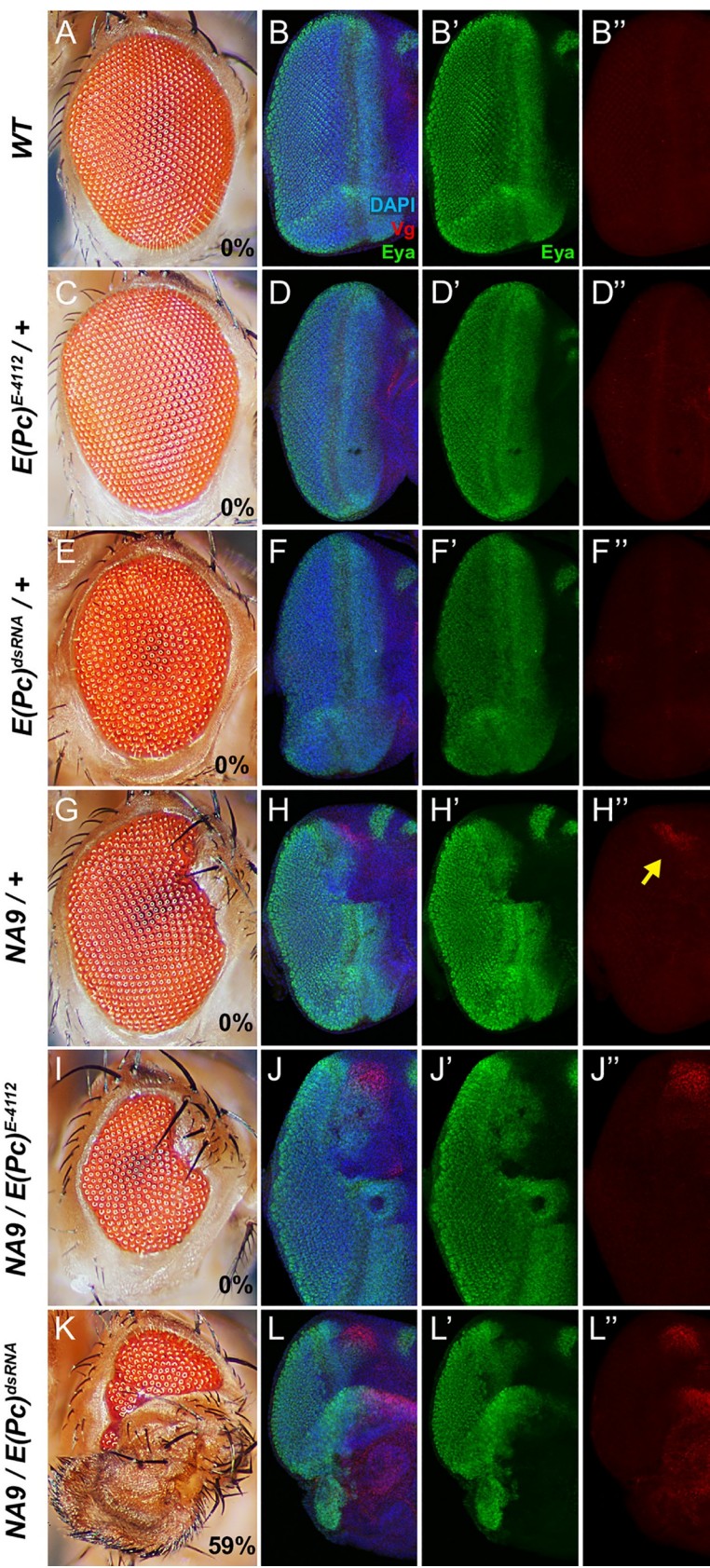

**Fig 6. *E(Pc)* inactivation enhances *NA9*-induced eye-to-wing transdetermination. (A, C, E, G, I, K)** Micrographs of adult *Drosophila* eyes. **(B, D, F, H, J, L)** Third instar larval eye discs immunostained with anti-Vestigial (Vg) and anti-Eyes Absent (Eya) antibodies. DAPI staining marks the nuclei. The following genotypes were analyzed: (A, B) *ey-Gal4/+* referred to as *WT*. (C, D) *ey-Gal4/E(Pc)*^E-4112^. (E, F) *ey-Gal4/UAS-E(Pc)*^dsRNA^. (G, H) *ey-Gal4, UAS-NA9/+*. (I, J) *ey-Gal4, UAS-NA9/E(Pc)*^E-4112^. (K, L) *ey-Gal4, UAS-NA9/UAS-E(Pc)*^dsRNA^. Proportion (%) of eyes exhibiting ectopic wing material is indicated at the bottom right of each eye micrograph. Quantifications are shown in Table 2. Yellow arrow points to the anterior dorsal margin exhibiting Vg expression.

ectopic wing structures. This observation is consistent with a close functional relationship between NA9 and CBP in this event.

CBP/p300 enzymes work as epigenetic regulators owing to their ability to acetylate specific lysine residues on distinct histones, which in turn play critical roles in the transcriptional activation of target genes [63,64]. Counteracting this activity are histone deacetylases (HDACs) that catalyze the removal of acetyl groups on specific lysine residues [65]. If NA9 is impeding Nej activity, we reasoned that reducing the dose of *Drosophila* HDACs might suppress NA9 eye phenotype. The fact that NA9 also associates with HDAC1 in human cells provide further interest in assessing for genetic interaction between NA9 and HDAC enzymes. *Drosophila* express five distinct class I/II *HDAC* genes among which *HDAC1* (aka *Rpd3*) appears to be a major contributor to histone deacetylation [66]. Supporting our hypothesis, *HDAC1* knockdown strongly suppressed *NA9*-induced eye phenotype as well as Vg misexpression (Fig 7G and 7H and Table 2). In contrast, *HDAC1* overexpression enhanced NA9 eye phenotype and Vg expression in the eye (Fig 7I and 7J and Table 2). Together, these observations strongly suggest that the ability of NA9 to influence eye cell fate is related to histone acetylation.

## Various *Enhancers of NA9* exacerbate NA9 transdetermination activity

In addition to *E(Pc)*, we wondered whether other genetic modifiers identified in the screen influenced NA9 transdetermination activity. To address this question, we assessed the ability of four groups of *Enhancers of NA9* to impinge on Vg levels when introduced in *NA9*-expressing eye discs. In an otherwise WT background, our reference *UAS-NA9* transgenic line (line 5) driven by *ey-Gal4* leads to 67% of eye discs with Vg expression (S11A Fig and Table 2). Markedly, crossing in heterozygous mutations for *emb*, *gpp*, *Rae1* and *CG6583* significantly enhanced the number of eye discs exhibiting Vg expression (S11B–S11E Fig and Table 2). Furthermore, the expression areas were consistently larger (S11B–S11E Fig). These observations suggest that, in addition to *E(Pc)*, other *Enhancers of NA9* from the screen impinge on NA9 transdetermination activity.

## Discussion

In this study, we present the development and characterization of a *Drosophila* eye phenotype designed to report on the activity of the NA9 oncoprotein and its use in a screen to identify genetic modifiers. Results from the screen revealed the ability of NA9 to impinge on epigenetic regulation and in turn alter cell fate determination. Moreover, they suggest that various intracellular functions are perturbed by NA9, which might reflect some of the pre-leukemic conditions prevailing in *NA9*-expressing bone marrow cells.

Expression of *NA9* with *ey-Gal4* perturbed eye development (Fig 1). The disruptive property of NA9 depended on the same functional elements as those initially defined in mammals, namely, the NUP98 portion, the HOXA9 homeodomain as well as the adjacent PBX-interaction motif (PIM) (Fig 1). These observations indicated that DNA binding and interaction with a PBX-like protein are critical for the effects of NA9 in the eye. Moreover, the NUP98 portion also probably comes into contact with resident proteins, thus impacting their physiological functions. Consistent with the involvement of a PBX-like protein, knockdowns of the

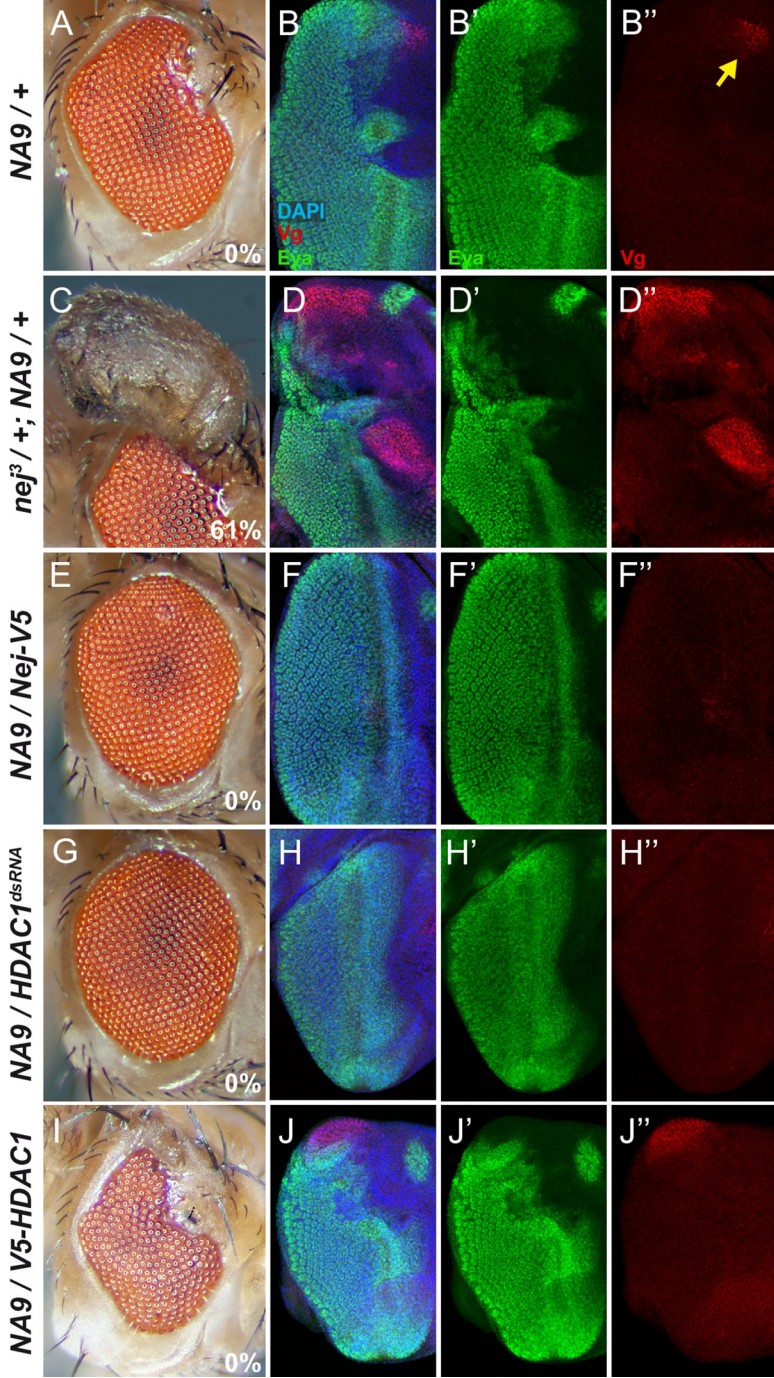

**Fig 7.** *Drosophila CBP/p300* **and** *HDAC1* **activity influence** *NA9*-**mediated transdetermination. (A, C, E, G, I)** Micrographs of adult *Drosophila* eyes. **(B, D, F, H, J)** Third instar larval eye discs immunostained with anti-Vestigial (Vg) and anti-Eyes Absent (Eya) antibodies. DAPI staining marks the nuclei. The following genotypes were analyzed: (A, B) *ey-Gal4, UAS-NA9/+*. (C, D) *nej³/+; ey-Gal4, UAS-NA9/+*. (E, F) *ey-Gal4, UAS-NA9/+; UAS-Nej-V5/+*. (G, H) *ey-Gal4, UAS-NA9/UAS-HDAC1^{dsRNA}*. (I, J) *ey-Gal4, UAS-NA9/+; UAS-V5-HDAC1/+*. Proportion (%) of eyes exhibiting ectopic wing material is indicated at the bottom right of each eye micrograph. Quantifications are shown in Table 2. Yellow arrow points to the anterior dorsal margin exhibiting Vg expression.

*Drosophila* TALE factors EXD or HTH also suppressed the NA9 eye phenotype (Fig 1). Conversely, co-expression of NA9 and HTH exhibited functional collaboration by promoting cell proliferation (Fig 2). The latter phenomenon is reminiscent of the cooperation observed between NA9 and the HTH homolog MEIS1 in AML [26]. Given these parallels with mammalian models, we conducted a screen for modifiers of NA9 on the basis that it could identify genetic modifiers whose mammalian homologs are relevant for NA9 leukemogenic activity.

The screen delineated 29 complementation groups of modifiers, 16 of which were assigned to a specific locus. Impressively, the genes identified encompass a variety of cellular processes such as epigenetic regulation, nuclear export, translation, cytoskeletal organization, cell polarity, and epithelial morphogenesis. These functions may reflect the wide range of effects of NA9 on expressing cells and their microenvironment. Among the recovered modifiers, we isolated mutations in three genes encoding fly counterparts of factors known to contribute to NA9 activity in mammals, namely, the transcription factor HTH, and the nucleocytoplasmic transporters EMB and RAE1. As mentioned above, MEIS1, the mammalian homolog of HTH, significantly accelerates the development of AML in *NA9*-expressing mice [26] and its gene has been identified as a common integration site in *NA9*-expressing BXH2 mice [67]. With respect to EMB (XPO1 in mammals) and RAE1, their mammalian homologs have been shown to contribute to NA9-induced leukemia through their association with the FG / GLFG motifs and GLEBS domain of NA9, respectively, which appears to interfere with nucleocytoplasmic transport [29–31,41,42]. Our work provides an unbiased genetic demonstration of a role for EMB and RAE1 as modulators of NA9 function. Together, these results confirm the ability of our approach to identify *bona fide* NA9 functional partners.

## NA9 influences epigenetic regulation

Recent studies in mammals have suggested a role for NA9 in chromatin remodelling by modulating epigenetic regulation. In particular, NA9 has been found to collaborate with epigenetic factors and transcriptional regulators such as MLL, p300/CBP and HDAC1 to influence target gene expression [32,33,61]. Consistent with these findings, the screen identified four epigenetic modulators: the Tip60 subunit E(Pc), the H3K79 methyltransferase Grappa (Gpp), the CP2 transcription factor Grainyhead (Grh) and the SERTAD protein Taranis (Tara) (Table 1). In addition, we found that NA9 genetically interacts with other Tip60 subunits (Fig 5) as well as with Nej and Rpd3, the *Drosophila* homologs of p300/CBP and HDAC1, respectively (Fig 7). These last two genetic interactions are consistent with previous work in mammals, which reported a physical contact between NA9 and p300/CBP or HDAC1 [27,28].

E(Pc) was originally identified as an atypical member of the repressive Polycomb group (PcG) proteins because its loss-of-function did not lead to homeotic transformation as normally observed for other PcG factors, but it nevertheless enhanced PcG mutations [68]. It was therefore considered early on as a transcriptional repressor. On this basis, it is intriguing that mutations in *nej*, which encodes a general transcriptional activator, behaved similarly to *E(Pc)* alleles and enhance NA9 activity. The function of E(Pc) has proven to be more complex than originally thought. Indeed, E(Pc) and homologous proteins in other species have been shown to be part of the Tip60/NuA4 multi-subunit complex that, in a context-dependent manner, act either as an activator or repressor of transcription, primarily through its HAT activity towards histones H2A and H4 as well as towards specific nuclear factors [55,69,70]. Therefore, it may well be that in Drosophila, E(Pc), through the Tip60/NuA4 complex, and Nej control a set of functionally related genes that are critical for eye fate commitment and that their dysregulation upon *E(Pc)* or *nej* loss-of-functions enhances the eye-to-wing transdetermination event induced by NA9 expression.

Markedly, EPC1/2 and DOT1L, the human homologs of E(Pc) and Gpp, respectively, have been previously described as co-factors in MLL-induced leukemia [71,72]. Moreover, the Grh-like GRHL2 protein has been shown to inhibit the histone acetyltransferase activity of p300 [73] as well as to associate physically with MLL3 and MLL4 in epithelial cells [74]. The inhibitory function of GRHL2 towards p300 may explain why *grh* alleles were recovered as *NA9* suppressors. Furthermore, since MLL has been found to physically interact with NA9 and to be necessary for its leukemogenic activity, functional interactions of NA9 with EPC1/2, DOT1L and GRHL could be mediated by MLL [32,33].

Finally, Tara has previously been identified as a mediator of the repressive (PcG) and activating Trithorax Group (TrxG) proteins [75]. A link to the cell cycle machinery has also been described [76]. The mammalian Tara homolog SERTAD2 is overexpressed in multiple human tumors and its protein levels are regulated by proteasomal degradation following XPO1-mediated nuclear export [77]. As NA9 activity is related to XPO1 function, Tara levels might be affected by NA9.

## A role in cell polarity / basement membrane organization

Intriguingly, the NA9-dependent screen also identified several genes coding for cell polarity factors such as the Scribble complex protein Lethal (2) giant larvae (L(2)gl) [78], the atypical cadherin Stary night (Stan; aka Flamingo) [79] and the immunoglobulin domain-containing cell adhesion molecule Echinoid (Ed) [80]. A connection between Stan, Ed and Grh has previously been demonstrated in *Drosophila*. In wing imaginal discs, Grh participates in cell polarity by controlling Stan's expression [81]. Moreover, Ed has been shown to regulate Stan's endocytosis in inter-ommatidial cells of the eye that in turn controls the rotation of ommatidial clusters [82]. The *NA9* screen has also identified *short stop* (*shot*) that encodes the *Drosophila* homolog of Dystonin, which links microtubules and the actin cytoskeleton in flies and vertebrates [83]. Interestingly, *shot* interacts genetically with *stan* and regulates planar cell polarity in the wing [84]. Although the underlying mechanism linking NA9 to PCP proteins is currently unknown, the interaction between CELSR2 (Stan's mammalian counterpart) and Frizzled-8 (Wnt/Wg pathway ligand) appears to be necessary in the hematopoietic niche for the maintenance of long-term HSCs [85]. In addition, recent data suggests the involvement of the Scribble complex in the regulation of HSC biology with a potential role in AML [86,87]. Interestingly, we have also identified the matrix metalloproteinase Mmp2, an extracellular protease responsible for extracellular matrix (ECM) degradation. Its mammalian counterpart MMP15 (aka MT2-MMP) is apparently expressed in several myeloid cell lines and in AML samples, which suggests a contribution to the invasiveness properties of NA9-mediated leukemia [88,89].

## A role in translation

Finally, the screen recovered mutations in translation-related genes such as *eIF3b* and *eIF3i*, which encode subunits of the eIF3 translation initiation complex, and the asparaginyl-tRNA-synthetase, *AsnRS*. A recent protein screen for HOXA9 interactors identified several subunits of the eIF3 complex [53]. Additionally, several subunits of translation initiation complexes were found to be involved in AML, such as eIF4E and different subunits of the eIF3 complex (e.g., eIF3B, eIF3D and eIF3K) [90–92]. Together, these findings support the idea that translational modulation is another means by which NA9 influences the development of AML.

## NA9 activity promotes cell fate switch in the Drosophila eye disc

The diversity of gene functions identified in the screen suggests that NA9 perturbs several cellular mechanisms that together produce the adult eye phenotype. An open question is whether

the phenotype results from pleiotropic defects or rather from a specific developmental disturbance. Our characterization of the phenotype supports the second scenario.

By inspecting the expression pattern of three key morphogens involved in eye development (Hh, Wg and Dpp), we initially found that *NA9* expression under *ey-Gal4* had no obvious impact on *hh* expression, but altered the expression of *wg* and *dpp*. Intriguingly, the effects were limited to the dorsal part of the eye disc (S1 Fig). On the one hand, *NA9* increased Wg protein levels (similar effects on *wg* expression; not shown) at the anterior margin of the dorsal area. On the other hand, it reduced *dpp* expression in the dorsal area of the morphogenetic furrow (MF) between the margin and the equator, but leaving it unaffected at the dorsal margin (S1 Fig). As *dpp* expression is necessary for MF progression, its reduced expression probably explains the MF progression delay observed on the dorsal side (S1 Fig).

How NA9 impinges on *wg* and *dpp* expression is currently not known. While further studies will be required to address this issue, it appears that NA9 does not work alone in this event and that it involves cell fate switch. Indeed, since NA9 does not alter *wg* and *dpp* expression on the ventral compartment of third instar eye discs, it indicates that it is not sufficient *per se* to change their expression, but that some other factors/conditions restricted to the dorsal side are also involved. Moreover, we found that the *ey-Gal4* driver was inactive in the presence of NA9 in the anterior dorsal area of third instar eye discs (the region that exhibits elevated Wg levels in S1 Fig), although a lineage tracing experiment demonstrated *ey-Gal4* activity at earlier time points in this area (S9 Fig). This suggests that the *NA9* transgene was expressed at earlier stage of eye disc development and that NA9 activity precluded *eyeless*-driven activity. Consistent with this model, we observed a repression of retinal determination gene expression such as *eyes absent* in the anterior dorsal compartment of third instar eye discs (Fig 6). Finally, by characterizing a genetic interaction between *NA9* and *E(Pc)*, we found that *NA9* expression induces cells at the margin of the anterior dorsal area to adopt a wing fate as revealed by Vg and Sd staining (Figs 6 and S7).

A number of studies have reported the presence of "weak points" within imaginal discs, which correspond to small groups of cells prone to enhanced cell proliferation and cell fate switch caused by derepressed selector gene expression [93]. These areas are linked to imaginal disc regeneration following tissue injury and appear to be epigenetically reprogrammable [93,94]. They have been located to specific sub-regions of imaginal discs including the anterior dorsal area of the eye disc [95]. Experiments conducted in leg discs have demonstrated the concomitant requirement for Wg and Dpp signaling in the induction of *vg* expression and leg-to-wing transformation [96]. Altered expression of PcG and TrxG genes in weak points have also been reported to induce transdetermination [57,97,98].

Given the ability of NA9 to genetically and physically interact with various epigenetic regulators, *NA9* expression during eye development could alter the epigenetic state of cells within weak points leading to their reprogramming to the wing fate. For example, previous studies have reported the presence of Polycomb Responsive Elements (PREs) within the *wg* and *vg* genes. PREs act as silencers of gene expression upon assembling PcG protein complexes [99]. Disruption of normal PRE function has been shown to induce *wg* and *vg* expression in leg discs [98]. By interfering with specific epigenetic regulators, *NA9* might impede the function of the *wg* and *vg* PREs, thus inducing their expression.

The mechanistic basis underlying transdetermination in flies and in mammals remains poorly understood. Its connection to tissue regeneration and cell plasticity is fascinating and underscores the great potential of a better characterization. In addition to providing an unprecedented list of functional modulators of NA9 activity that could be used as starting points for new studies in NA9-mediated AML models in mammals, our work offers a powerful system and several associated genes for further investigating transdetermination.

## Materials and methods

### Fly stocks and husbandry

Flies were kept on standard cornmeal-based or semi-defined medium (https://bdsc.indiana.edu/information/recipes/index.html). Details on fly stocks, medium or rearing temperatures for each figure are available in S1 Table.

FLP-out clones were induced 72 hr after egg deposition by a 20 min heat shock at 38˚C. Third instar larval eye discs were dissected 72 hr after heat shocks.

### Cloning and transgenic flies

The human *NUP98-HOXA9* (*NA9*) cDNA [25] was cloned downstream to the *ey-hsp70* enhancer-promoter cassette in the *pCasper P*-element vector. Standard molecular biology procedures were used to generate the *NA9* variants, namely, $NUP98^{\Delta CT}$ (amino acids 1–469), $HOXA9^{\Delta NT}$ (amino acids 163–271), $NA9^{HD}$ (N562A) and $NA9^{PIM}$ (W506A) cDNAs. These were then cloned into the *pUAST P*-element vector [100]. The *P*-element constructs were introduced into the $w^{1118}$ fly stock by *P*-element-mediated germline transformation as described previously [101].

### NA9-dependent screen, allele mapping and sequencing

For mutagenesis, $w^{1118}$ males isogenized for the second and third chromosomes ($w^{1118};iso2;3$) were fed with 25 mM ethyl methanesulfonate, 100 mM Tris-HCl pH 7.5 and 10% sucrose. Mutagenized males were then mated with either *adv/CyO*, *P[w⁺*, *ey-NA9]* (*CNA9*) or *e*, *ftz*, *ry /TM3*, *P[w⁺*, *ey-NA9]* (*TNA9*) virgin females. F1 progeny was scored for modification of eye size and shape compared to the parental line using a MZ8 stereomicroscope (Leica). Complementation tests based on recessive lethality were conducted to establish allelism among the modifiers linked to chromosome 2 or 3. Complementation groups were mapped using the DrosDel (v3.1) deficiency kit collection. Allelism to specific loci within cytological positions was determined using available recessive lethal alleles from stock centers. Molecular confirmation of a subset of recovered loci was provided by exome sequencing of mutant alleles as previously described [102].

### GFP fluorescence intensity measurement

GFP-positive third instar eye-antennal discs were dissected in ESF921 medium (Expression systems) and transferred to 1X phosphate-buffered saline (PBS 1X), 5% Bovine Serum Albumin (BSA). Unfixed discs were then mounted in Mowiol (Sigma) and imaged immediately using an Axio Imager microscope equipped with a 20X objective and a GFP filter (Zeiss). The mean GFP intensity in eye discs was calculated using the Adobe Photoshop software. Briefly, using the rectangle selection tool, a 400x150 pixels box was drawn in the posterior region of eye discs. The mean Gray Value (measurement of brightness) comprised within the box was then determined using the Photoshop Measurement feature. Five eye discs per genotype in triplicate were analyzed and the GFP intensity values were normalized to control discs.

### Western blotting

To prepare whole larvae lysates, 15 third instar larvae were washed once in PBS 1X with 0.2% Triton X-100 (PBT 0.2%) and once in PBS 1X. Larvae were then put in 200 μl of RIPA lysis buffer (50 mM Tris at pH 8.0, 150 mM NaCl, 1% NP-40, 0.5% sodium deoxycholate, 0.1% SDS, 1 mM PMSF, 1 mM $Na_3VO_4$, 10 μg/ml aprotinin, 10 μg/ml leupeptin) and protein were

extracted using a pellet pestle adapted for 1.5 ml Eppendorf tube. After 15 min incubation on ice, debris were removed by centrifugation at 12,000g for 15 min at 4˚C.

Lysates were resolved on SDS-PAGE and transferred to nitrocellulose membranes. Blotted proteins were immuno-detected using rabbit anti-HOXA9 (1:2000, Milipore Sigma), rabbit anti-NUP98 (1:1000, Cell Signalling), mouse anti-β-Gal (1:2000, Fisher Scientific), and mouse anti-Actin (1:1000, Boehringer Manheim) antibodies.

## Compound treatment of developing flies

Crosses were set up in FACS tube (Falcon *#352235)* containing 1 ml of semi-defined medium supplemented with DMSO or 6.25 μM of selinexor (Selleckchem S7252, 50 mM stock solution) at 25˚C.

## Immunostaining and imaging

For immunostaining of eye-antennal imaginal discs, third instar larvae were dissected in ESF921 medium (Expression Systems) supplemented with 1 mM $CaCl_2$, then fixed for 15 min in 4% paraformaldehyde, 1 mM $CaCl_2$, PBS 1X and washed three times in PBS 1X, 0.2% Triton X-100 (PBT 0.2%). Primary antibody sources and dilutions were as follow: mouse anti-Eya (1:200, DSHB), rabbit anti-Vg (1:1000, kind gift of K. Guss and S. Carroll [103]), mouse anti-Wg (1:200, DSHB), rat anti-Elav (1:200, DSHB), mouse anti-βGal (1:2000, Fischer Scientific), rabbit anti-Hth (1:500, kind gift of A. Salzberg [104]), mouse anti-Exd (1:100, DSHB), Rabbit anti-Emb (1:200, kind gift of C. Samakovlis, [105]), rat anti-Ed (1:1000, kind gift of L. Nilson, [106]), mouse anti-Stan (1:100, DSHB), rabbit anti-Grh (1:200, kind gift of W. McGinnis), mouse anti-Shot (1:200, DSHB), rabbit anti-HDAC1 (1:1000, Abcam), and mouse anti-V5 epitope (1:1000, ThermoFischer Scientific). Eye-antennal discs were incubated with primary antibodies diluted in PBT 0.2%, 5% BSA overnight at 4˚C. Tissues were washed three times in PBT 0.2% and incubated at room temperature for 2 hours with appropriate species-specific fluorophore-conjugated secondary antibodies (1:1000, Molecular Probes) diluted in PBT 0.2%. Eye-antennal discs were then washed once in PBT 0.2% with 100 ng/ml DAPI (Sigma), twice in PBT 0.2% and mounted in Mowiol (Sigma). Imaging was performed using a Zeiss LSM510 or LSM700 confocal microscopes equipped with a 40X objective.

For BrdU incorporation, third instar eye-antennal imaginal discs were dissected in ESF921 medium and incubated with BrdU (1 μg/ml) for 30 minutes at 25˚C, fixed for 15 min in 4% paraformaldehyde, PBS 1X. Discs were then washed three times in PBT 0.3%, incubated in PBS 0.3%, 2N HCl for 30 min at room temperature, blocked for 15 min in PBT 0.3%, BSA 2% and incubated with mouse anti-BrdU (BD Pharmingen 555627, 1:200). Discs were washed three times in PBT 0.3% and incubated 2h with species-specific secondary antibodies at room temperature (Cy3-conjugated anti-mouse for α-BrdU). Discs were then washed once with PBT 0.3%, 100 ng/ml DAPI and twice in PBT 0.3%, and mounted in Mowiol (Sigma). Imaging was performed using a Zeiss LSM510 confocal microscope equipped with a 40X objective.

For adult eye imaging, flies were collected and frozen at -80˚C for at least 2 hrs. Eye imaging was conducted using a stereomicroscope (Leica) equipped with a camera (Nikon). Five pictures were taken per eye from the apical to basal focal plan and 3D reconstruction of the eye was made using the Helicon Focus software (HeliconSoft).

Tissue overgrowth was measured in adult eyes only. Measurements were conducted by outlining the overgrowths using the Lasso tool in Photoshop. The areas (square pixels) were then determined using the Measurement log. The average size of tissue overgrowths produced by NA9 and HTH co-expression was normalized to 1 and used for comparison to other genotypes. The area value was set to 0 when no overgrowth was detected. Eye size was measured in

a similar manner, but in this case, the average size of control adult eyes (*LacZ*) was normalized to 1. Original data and statistics are reported in the S1 Data.

### RT-qPCR analysis

For RT-qPCR analysis, 15 to 20 eye discs were dissected and total RNA was extracted using the RNeasy Micro kit (Qiagen) according to the manufacturer's procedures. Reverse transcription was performed on 200 ng of total RNA using the High capacity reverse transcription kit from Applied Biosystems. Universal Probe Library design center (Roche) was used to design Taqman qPCR assays. Reactions were performed with the TaqMan Real-Time PCR Master Mix and analyzed with the ViiA 7 Real-Time PCR System. Primers used for RT-qPCR analysis are listed in S2 Table.

## Supporting information

**S1 Fig. Effects of NA9 on Wingless (Wg), *hedgehog* (*hh*) and *decapentaplegic* (*dpp*) expression during eye development.** Third instar larval eye discs immunostained with **(A, B)** anti-Wingless (Wg) and anti-Elav (marks photoreceptor neurons). **(C, D)** Anti-βGal stainings (red) report on *dpp* expression. **(E, F)** Anti-βGal stainings (red) report on *hh* expression. DAPI stainings mark the nuclei. GFP delineates the areas of *ey-Gal4* activity. The following genotypes were analyzed: (A) *ey-Gal4/+; UAS-GFP.nls/+* referred to as *CTL*. (B) *ey-Gal4/UAS-NA9; UAS-GFP.nls/+* referred to as *NA9*. (C) *ey-Gal4/dpp-LacZ* referred to as *CTL*. (D) *ey-Gal4, UAS-NA9/dpp-LacZ* referred to as *NA9*. (E) *ey-Gal4/+; hh-LacZ/+* referred to as *CTL*. (F) *ey-Gal4, UAS-NA9/+; hh-LacZ/+* referred to as *NA9*.
(TIF)

**S2 Fig. *NA9* variant constructs are expressed to similar levels.** Immunoblots monitoring protein levels from whole larval extracts for the different NA9 variants used in this study. The *ey-Gal4* line was used to drive the expression of *UAS-LacZ* or the following *UAS* constructs: *UAS-NA9* (WT), *UAS-NA9$^{HD}$* (HD), *UAS-NA9$^{PIM}$* (PIM), *UAS-HOXA9$^{\Delta NT}$* (HOX) or *UAS-NUP98$^{\Delta CT}$* (NUP). Protein levels were assessed using antibodies against HOXA9 or NUP98. βGal levels were determined as control for *ey-GAL4* activity. Actin levels were used as loading control. *CTL* corresponds to cell extracts made from the control *w$^{1118}$* line.
(TIF)

**S3 Fig. The eye phenotype induced by *NA9* expression is dosage sensitive. (A-D)** Micrographs of adult *Drosophila* eyes of the following genotypes: (A) *ey-Gal4/+; UAS-GFP/+* referred to as *CTL*. (B-D) *ey-Gal4, UAS-NA9/+; UAS-GFP/+* referred to as *NA9*. Flies were raised at 18˚C, 25˚C or 29˚C as indicated. Mean eye size (expressed as percent compared to *CTL*), is indicated at the bottom right of each eye micrograph. Five flies were quantified per condition. **(E)** Quantification of GFP fluorescence in eye imaginal discs. Expression of the *NA9* transgene is enhanced by temperature elevation during development, which correlates with enhanced phenotypic strength in adult eyes. Stars denote statistically significant variations ($p \leq 0.05$, Student's *t*-test) in GFP fluorescence compared to conditions at 25˚C.
(TIF)

**S4 Fig. Effect of heterozygous alleles on transgene expression. (A)** Third instar eye imaginal disc expressing *UAS-GFP.nls* under *ey-Gal4*. Mean GFP fluorescence intensity was used as a proxy for assessing variation in *ey-Gal4* activity. GFP fluorescence was quantified in the posterior region of eye discs (white box) using the Photoshop Measurement feature. The following genotypes were analyzed: **(B)** *ey-Gal4/+; UAS-GFP.nls/suppressor alleles* and **(C)** *ey-Gal4/+; UAS-GFP.nls/enhancer alleles* as indicated on the panels. Fluorescence intensities are

normalized to CTL (*ey-Gal4/+; UAS-GFP.nls/+*). Error bars represent standard deviations (SD) from at least three independent experiments. The green and red stars highlight a statistically significant ($p \leq 0.05$, Student's *t*-test) increase or decrease in GFP fluorescence, respectively. **(D)** Amino acid sequence alignment of PBX-interacting regions from *Drosophila* HTH and human MEIS-related homeobox proteins. Identical amino acids are shown in red, whereas similar amino acids are shown in blue. The position of the amino acid change (K180N) found in the *hth*[S-195] allele is shown on top of the HTH sequence.
(TIF)

**S5 Fig. Depletion of candidate genes by dsRNA during eye development. (A-L)** Micrographs of representative adult *Drosophila* eyes of (A) *ey-Gal4/+* referred to as *CTL* or (B-L) *ey-Gal4/*specific *UAS-dsRNA* constructs as indicated on the panels.
(TIF)

**S6 Fig. Validation of dsRNA and cDNA expression constructs used in this study. (A-J)** Third instar larval eye discs were immunostained with the indicated antibodies (red) to monitor knockdown efficiency of (A-H) dsRNA lines or expression of (I, J) V5-tagged cDNAs as indicated to the left of each panel. DAPI staining marks the nuclei, whereas GFP fluorescence identifies the areas of transgene expression. The *flp-out* line (*hs-flp; Act5C > CD8 > GAL4, UAS-GFP*) was used to clonally induce the expression of GFP and the indicated *UAS* constructs as single copies. When antibodies were not available to assess knockdown efficiency, qPCR analysis were performed instead using mRNA transcripts isolated from eye discs **(K-O)** or lymph glands **(P)**. The mean RQ values of at least two independent experiments are shown. Statistical significance was determined using a Student's *t*-test.
(TIF)

**S7 Fig. NA9 induces *scalloped* (*sd*) expression during eye development.** Third instar larval eye discs immunostained with anti-βGal as a reporter for *sd-LacZ* expression. The following genotypes were analyzed: **(A)** *sd-LacZ/+; ey-Gal4/+* referred to as *CTL*. **(B)** *sd-LacZ/+; ey-Gal4/UAS-NA9* referred to as *NA9*. **(C)** *sd-LacZ/+; ey-Gal4/UAS-NA9*[line6] referred to as *NA9*[line6]. DAPI staining marks the nuclei.
(TIF)

**S8 Fig. A stronger *NA9* expression line further promotes eye-to-wing transdetermination. (A, C, E)** Micrographs of adult *Drosophila* eyes. **(B, D, F)** Third instar larval eye discs immunostained with anti-Vestigial (Vg) and anti-Eyes Absent (Eya) antibodies. DAPI staining marks the nuclei, whereas GFP defines the areas of transgene expression. The following genotypes were analyzed: (A, B) *ey-Gal4/+; UAS-GFP.nls/+* referred to as *CTL*. (C, D) *ey-Gal4/ UAS-NA9; UAS-GFP.nls/+* referred to as *NA9*. (E, F) *ey-Gal4/+; UAS-GFP.nls/UAS-NA9*[line6] referred to as *NA9*[line6]. The proportion (%) of eyes presenting ectopic wing formation is indicated at the bottom right of adult eye micrographs. Quantifications are shown in Table 2. Transgene expression of *UAS-NA9*[line6] is approximately 6-fold higher compared to the *UAS-NA9* used in this study and previously referred to as line 5 [35].
(TIF)

**S9 Fig. Lineage tracing of *NA9* transgene expression driven by *ey-Gal4*.** Real-time expression (RFP) and lineage expression (GFP) was determined for the following genotypes: **(A)** *ey-Gal4/+; G-Trace/+* referred to as *CTL*. **(B)** *ey-Gal4, NA9, G-Trace* referred to as *NA9*. DAPI staining marks the nuclei. *G-Trace* refers to *UAS-RFP, UAS-FLP, Ubi-p63E(FRT.STOP)nEGFP* [60].
(TIF)

**S10 Fig. Modulation of Nej or HDAC1 activity does not induce eye-to-wing transdetermination. (A, C, E, G, I)** Micrographs of adult *Drosophila* eyes. **(B, D, F, H, J)** Third instar larval eye discs immunostained with anti-Vestigial (Vg) and anti-Eyes Absent (Eya) antibodies. DAPI staining marks the nuclei. The following genotypes were analyzed: (A, B) *ey-Gal4, UAS-GFP/+* referred to as *CTL*. (C, D) *nej³/+; ey-Gal4, UAS-GFP/+*. (E, F) *ey-Gal4, UAS-GFP/ UAS-HDAC1^{dsRNA}*. (G, H) *ey-Gal4, UAS-GFP/UAS-nej-V5*. (I, J) *ey-Gal4, UAS-GFP/ UAS-V5-HDAC1*. Proportion (%) of eyes exhibiting ectopic wing material is indicated at the bottom right of each eye micrograph. Quantifications are shown in Table 2.
(TIF)

**S11 Fig. *Enhancers of NA9* recovered from the screen impinge on *NA9*-induced eye-to-wing transdetermination. (A-E)** Third instar larval eye discs immunostained with anti-Vestigial (Vg) and anti-Eyes Absent (Eya) antibodies. DAPI staining marks the nuclei. The analyzed genotypes are indicated to the left of the panels.
(TIF)

**S1 Table. Fly stocks, rearing conditions and antibodies used in this study.**
(XLSX)

**S2 Table. qPCR primers used in this study.**
(XLSX)

**S1 Data. Numerical data and statistics reported in this study.**
(XLSX)

## Acknowledgments

We are grateful to U. Banerjee, S. Carroll, B. Edgar, S. Gosh, K. Guss, A. Hermann, R. Mann, W. McGinnis, L. Nilson, A. Salzberg, C. Samakovlis, and V. Tsarouhas for providing fly stocks and reagents. We also thank the Bloomington, the VDRC and the National Institute of Genetics (Japan) stock centers for fly stocks and the Drosophila Genomic Resource Center for *P-element* vectors.

## Author Contributions

**Conceptualization:** Gwenaëlle Gavory, Caroline Baril, Gino Laberge, Guy Sauvageau, Marc Therrien.

**Data curation:** Gwenaëlle Gavory, Caroline Baril, Gino Laberge, Gawa Bidla, Surapong Koonpaew, Thomas Sonea, Marc Therrien.

**Formal analysis:** Gwenaëlle Gavory, Caroline Baril, Gino Laberge, Gawa Bidla, Surapong Koonpaew, Thomas Sonea.

**Funding acquisition:** Marc Therrien.

**Investigation:** Gwenaëlle Gavory, Caroline Baril, Gino Laberge, Gawa Bidla, Surapong Koonpaew, Thomas Sonea.

**Methodology:** Gwenaëlle Gavory, Caroline Baril.

**Project administration:** Caroline Baril, Gino Laberge, Guy Sauvageau, Marc Therrien.

**Supervision:** Caroline Baril, Gino Laberge, Guy Sauvageau, Marc Therrien.

**Validation:** Gwenaëlle Gavory, Caroline Baril, Gawa Bidla, Surapong Koonpaew.

**Writing – original draft:** Gwenaëlle Gavory, Caroline Baril, Marc Therrien.

**Writing – review & editing:** Gwenaëlle Gavory, Caroline Baril, Gino Laberge, Gawa Bidla, Surapong Koonpaew, Guy Sauvageau, Marc Therrien.

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
