## [Decision Letter · Decision Letter 0]

7 Apr 2021

Dear Marc,

Thank you very much for submitting your Research Article entitled 'A genetic screen in Drosophila uncovers the multifaceted properties of the NUP98-HOXA9 oncogene' to PLOS Genetics.

Two experts in the field have reviewed your manuscript, and I have read it as well. I am pleased to tell you that the manuscript is potentially suitable for publication in PLOS Genetics. However, the reviewers have comments and concerns that need to be addressed in a revised manuscript. You can read their reviews at the end of this email.

We therefore ask you to modify the manuscript according to the review recommendations. Your revisions should address the specific points made by each reviewer.

Yours sincerely,

David A. Wassarman

Guest Editor

PLOS Genetics

Peter McKinnon

Section Editor: Cancer Genetics

PLOS Genetics

Reviewer's Responses to Questions

**Comments to the Authors:**

Reviewer #1: The manuscript by Gavory and colleagues describes their efforts to use the Drosophila eye as a model to study acute myeloid leukemia (AML), a devastating human disorder. One inducer of AML is a translocation that fuses the N-terminal Phe-GLy repeats of NUP98 with the HOX9 transcription factor (ortholog of AbdA/B). Expression of this fusion (called NA9) in the Drosophila eye results in a small roughened eye. In this background the authors conducted a enhancer/suppressor screen and identified 29 complementation groups that modify the rough eye phenotype. A fraction of these are known chromatin modifiers. The authors focus on one type of modification – the transdermination of the eye into a wing. This phenomenon has been previously reported to occur in disc fragmentation/transplantation experiments, compound mutants in which Pax6 is down-regulated with other transcription factors, when the heterochromatin promoting gene winged-eye is over-expressed, and when Polycomb group membrs are downregulated.

The results in this paper are exciting for several reason and are appropriate for publication in PLoS Genetics. The authors were able to identify genes that are known to interact with NA9 in mammals thus validating the use of Drosophila as a model system. This, in and of itself, makes this study quite impactful. Second, the authors have uncovered a novel path towards transdetermining the eye into a wing. It appears that strong activation of Hox9/AbdAB targets along with modifications of epigenetic modifying factors are sufficient to switch the eye into a wing.

The paper is written very well, the figures are of the highest quality, and the data supports the conclusions. I would also say that the paper is very comprehensive. I only have few suggestions but overall I really like this manuscript.

1. I might have missed it but I think it would be a good control for the authors to deplete Vg and Scalloped (independently) and show that their loss suppresses the eye to wing transdetermination event.

2. It is interesting that depleting genes involved in activation of gene expression (nej) and repression of gene expression (E(Pc)) both lead to the eye to wing swith. Can the others comment further on this in the discussion.

3. Could the authors conduct a small mini-screen where they over-express fly AbdA oir AbdB and deplete (with RNAi lines) some of the factors that were identified in their major screen to see if they can induce the eye to wing transformation. I think this would be very interesting no matter how it turns out.

Reviewer #2: Gavory and colleagues make good use of a genetic modifier screen in Drosophila to identify interactors of oncogenic NA9 fusion gene that is implicated in AML. Homologs of known modifiers of NA9 in leukemogenesis were identified, supporting the idea that modifiers found in Drosophila will apply to humans. The results suggest that NA9 perturbs several cellular functions. The manuscript is written well. The data are of good quality and most of the conclusions are supported by the data (exceptions described below). Appropriate controls are included such as those used to exclude the possibility that modifiers modify ey-GAL4 activity. The screen identified hth which was known to modify from direct testing, thus the data are internally consistent. The most interesting finding is that NA9 changed the fate of eye cells into wing disc cells. In principle, the questions being addressed here are suitable for publication in PLoS Genetics, but the analysis is mostly genetic interactions (although with various phenotypic outputs) and the reader is left with a gene list and not a mechanism. For example, the analysis that led to the finding that NA9 causes eye-to-wing transdetermination is superficial; the manuscript would be stronger if there was some indication of exactly how NA9 is doing this. Specific comments are:

1. The authors show that PIM and HD regions of NA9 are needed in fly, as they are for transformation in mammalian cells. But they have not shown that FG repeats of NUP98 are needed. In fact, HOX Delta NT without NUP98 reduced eye size on its own. Its effect is less pronounced than that of NA9 but is the difference statistically significant? In other words, what are the p values for the comparison between NA9 vs. PIM, HD, delta NT or delta CT mutants in Fig. 1H? Is the difference between NA9 and HOXA9 delta NT significant enough to support the conclusion that FG repeats matter? Otherwise, the conclusion that ‘these results indicate that the NA9 eye phenotype depends on the known functional elements of the oncoprotein’ is only partially supported by the data.

2. The rescue of the NA9 eye phenotype by Exd and hth RNAi in Fig. 1 is very nice. But I am having trouble seeing the synergistic effect of hth and NA9 on tissue overgrowth in Fig 2. Is overgrowth measured from adult eyes or eye discs? If the adult eye, is overgrowth being measured beyond the expected adult eye size and how? If in eye discs, is it the GFP area that is measured? If so, I cannot see that NA9 + hth in Fig. 2H shows overgrowth over hth or NA9 alone. If anything, there seems to be less GFP in Fig. 2H than in Fig. 2F. In any case, better description of exactly what is being measured is needed.

3. The authors identified a previously undocumented activity of NA9 in transdetermination, from eye to wing in this case. They also identified a number of genetic/epigenetic modifiers of this effect. These are very interesting data. But an obvious connection is not being made. A large body of literature on transdetermination in Drosophila implicates Wg as playing a central role and some of the work is cited by the authors. The authors see up-regulation of Wg when they overexpress NA9, and this effect is greater in the dorsal part, which is where they see transdetermination (Fig. S1). Therefore, they should test if Wg is needed for NA9-induced transdetermination. Conditional Wg alleles or conditional expression of inhibitors may be used. In any case, some insight into the mechanism of transdetermination would improve the manuscript.

**Have all data underlying the figures and results presented in the manuscript been provided?**

Reviewer #1: None

Reviewer #2: Yes

PLOS authors have the option to publish the peer review history of their article (what does this mean?). If published, this will include your full peer review and any attached files.

Reviewer #1: No

Reviewer #2: No

---

## [Decision Letter · Decision Letter 1]

20 Jul 2021

Dear Dr Therrien,

We are pleased to inform you that your manuscript entitled "A genetic screen in Drosophila uncovers the multifaceted properties of the NUP98-HOXA9 oncogene" has been editorially accepted for publication in PLOS Genetics. Congratulations!

Yours sincerely,

David A. Wassarman

Guest Editor

PLOS Genetics

Peter McKinnon

Section Editor: Cancer Genetics

PLOS Genetics

Comments from the reviewers (if applicable):

Reviewer's Responses to Questions

**Comments to the Authors:**

Reviewer #1: The authors have satisfied my concerns. I appreciate their efforts. I support the publication of the manuscript in PLOS Genetics.

Reviewer #2: The revised version of the manuscript by Gavory et al. addresses my concerns. The authors have also made a good-faith effort to address the role of Wg in fate transformation. I support the publication of the revised version in PLoS Genetics.

**Have all data underlying the figures and results presented in the manuscript been provided?**

Reviewer #1: Yes

Reviewer #2: Yes

PLOS authors have the option to publish the peer review history of their article (what does this mean?). If published, this will include your full peer review and any attached files.

Reviewer #1: No

Reviewer #2: No

**Data Deposition**

http://datadryad.org/submit?journalID=pgenetics&manu=PGENETICS-D-21-00368R1

**Press Queries**

---

## [Editor Report · Acceptance letter]

9 Aug 2021

PGENETICS-D-21-00368R1 

A genetic screen in Drosophila uncovers the multifaceted properties of the NUP98-HOXA9 oncogene 

Dear Dr Therrien, 

We are pleased to inform you that your manuscript entitled "A genetic screen in Drosophila uncovers the multifaceted properties of the NUP98-HOXA9 oncogene" has been formally accepted for publication in PLOS Genetics! Your manuscript is now with our production department and you will be notified of the publication date in due course.

With kind regards,

Livia Horvath

PLOS Genetics

On behalf of:
